

# Impact-based temporal clustering of multiple meteorological hazard types in southwestern Germany

Katharina Küpfer[1,2], Alexandre Tuel[3, †], and Michael Kunz[1,2]

[1]Institute of Meteorology and Climate Research Troposphere Research (IMKTRO), Karlsruhe Institute of Technology (KIT), Germany
[2]Center for Disaster Management and Risk Reduction Technology (CEDIM), Karlsruhe Institute of Technology (KIT), Germany
[3]Institute of Geography and Oeschger Center for Climate Change Research, University of Bern, Switzerland
[†]Now at Galeio, Paris, France

**Correspondence:** Katharina Küpfer (katharina.kuepfer@kit.edu)

**Abstract.** A series of multiple meteorological extreme events in close succession can lead to a substantial increase in total losses compared to randomly distributed events. In this study, different temporal clustering methods are applied to insurance loss data on southwestern Germany from 1986 to 2023 for the following hazards: windstorms, convective gusts, hail, as well as pluvial, fluvial and mixed flood events. We assess the timing and significance of seasonal clustering of single hazard types as
well as their serial combination by use of both a simple counting algorithm and the clustering metric Ripley's K. Results show that clusters of damaging hazards occur mainly during May–August. Although clustering is significant only for certain hazard types compared to a random process, clustering is robust for a combination of multiple hazard types, namely hail, mixed or pluvial floods and storms. This particular combination of hazard types is also associated with higher losses compared to their isolated occurrence. Clustering results also depend on the method of defining independent events (Peaks-over-Threshold with
flexible lengths vs. Hours Clause with fixed lengths) and their resulting duration. This study demonstrates the relevance of considering multiple hazard types when evaluating clustering of meteorological hazards.

## 1 Introduction

Weather- and climate-related hazards frequently cause considerable loss and damage in Germany, such as the extreme flood caused by the low pressure system *Bernd* in western Germany and Belgium in July 2021 (Mohr et al., 2023) or the storm
series *Dudley, Eunice*, and *Franklin* in February 2022 (Mühr et al., 2022). These events can also lead to fatalities and major heavy societal impacts, such as damage to critical infrastructure with potential long-term consequences (Schäfer et al., 2021). In Central Europe, the most relevant hazard types with regard to losses are hydro-meteorological extremes (European Environment Agency, 2023), such as floods, winter storms, hail, and convective gusts. Damage by those hazard has increased globally (Banerjee et al., 2024) and in Europe (Kron et al., 2019), which can partly be attributed to climate change.
Between 1950 and 2013, total losses caused by natural hazards amounted to more than €30 billion in Europe, with floods and storms being the major drivers (Kreibich et al., 2014). Germany, and the southwest in particular, has been a hotspot for



damaging meteorological hazards in recent decades, resulting in high losses compared to other regions in Europe (Kron et al., 2019). For example, the winter storm *Lothar* in 1999 caused total economic losses of more than €15 billion (Swiss Re, 2019), the hail event *Andreas* in 2013 led to an economic loss of more than €1 billion (Kunz et al., 2018), and most recently, in June
2024, southwestern Germany was hit by flooding leading to an expected economic loss of more than €2 billion (GDV, 2024).

These extreme events often do not occur in isolation. In southwestern Germany, for example, there was a sequence of multiple events in 2013: an exceptional flood occurred during the end of May and the beginning of June, which was followed by extreme heat and a severe hailstorm in July, as well as extreme heat in the beginning of August (Deutscher Wetterdienst (DWD), 2013). All of these events caused severe impacts, with inundated towns and villages after the flooding, damaged
roofs and facades of buildings due to large hail (Kunz et al., 2018), and blocked water routes due to drought hindering the transportation of goods (Thieken et al., 2016).

When a combination of multiple drivers and/or hazards contributes to societal or environmental risk, this is referred to as compound weather or climate events (Zscheischler et al., 2018). Similarly, the United Nations Office for Disaster Risk Reduction (UNDRR) defines multi-hazards as "the specific contexts where hazardous events may occur simultaneously, cascadingly
or cumulatively over time, and taking into account the potential interrelated effects" (UNDRR, 2016, p. 19). It has recently been shown that these compound or multi-hazard events are quite frequent: 19 % of events in the global disaster database EM-DAT can be classified as multi-hazard events, leading to an overproportionally high share of 59 % of global economic losses, with the primary meteorological hazard types being floods and storms (Lee et al., 2024). Despite this large proportion, risk models such as those used by the insurance industry generally consider the different hazard types independently (Hillier et al.,
2015; Mitchell-Wallace et al., 2017; Priestley et al., 2018). Risk analyses and risk management often lack this multi-hazard perspective as well (Kreibich et al., 2014).

Compound impacts can also occur, such as blocked traffic routes after a previous event (Mohr et al., 2023), disturbed emergency responses (Raymond et al., 2020), an increased recovery time (Ruiter et al., 2020), or additional damage to buildings after a storm if damaged roofs fail to stop rainwater from entering (Martius et al., 2016). With major damaging hazards
happening in close succession, this can lead to capacity problems for civil protection, local authorities, insurance companies and NGOs.

In the past years, many studies have investigated compound events, however, most of them focus on single hazard types, or multiple dry hazard types, such as heatwaves and droughts (Ruiter and Loon, 2022). There is only little research on the co-occurrence or compound occurrence of different hazard types, particularly in relation to their impact. However, this perspective
has gained increasing attention in recent years. For example, Ruiter et al. (2020) evaluate consecutive disasters of different types, and recent work focuses on the classification of multiple hazards including a range of hazard types (Claassen et al., 2023). Specifically, the interrelationship between flood and wind occurrence has received quite some attention (e.g., Hillier et al., 2015; Martius et al., 2016; Bloomfield et al., 2023, 2024; Hillier et al., 2024).

The temporal dependence of multiple hazards can be quantified using different clustering techniques. Temporal clustering,
also referred to as serial clustering (Mailier et al., 2006), has been widely investigated for single hazard types, mainly for extratropical cyclones in Europe (e.g., Vitolo et al., 2009; Mailier et al., 2006; Pinto et al., 2013, 2016; Karremann et al., 2014;





Dacre and Pinto, 2020), and heavy precipitation at various spatial scales (Barton et al., 2016; Kopp et al., 2021; Tuel and Martius, 2021b; Banfi and De Michele, 2024). There are also some studies on the serial clustering of hail (Barras et al., 2021) as well as droughts (Brunner and Stahl, 2023). The basic idea of serial clustering is to quantify the deviation of a binary time

series from a homogeneous Poisson process, i. e., a Poisson process with a constant rate of occurrence. This is typically done in two different ways. The dispersion index can identify overdispersion, i. e., the overall tendency of a time series to cluster (e.g., Mailier et al., 2006; Vitolo et al., 2009; Karremann et al., 2014). Another common approach is to use the Ripley's K function (e.g., Barton et al., 2016; Tuel and Martius, 2021a, b), which is a statistical metric that quantifies the average number of events around a random event in a time series.

To our knowledge, there has been no research on the identification and quantification of serial clustering considering multiple (meteorological) hazard types. We want to close this gap by using insurance loss data from a building insurance company in southwestern Germany. Although loss data only cover insured objects and do not necessarily represent societal damage, they are commonly used as a proxy, which allows to compare between different hazard types (Hillier et al., 2015). We use a counting method to identify clustering periods, and Ripley's K to assess the degree of clustering. Furthermore, we use two methods to

identify extreme events: a flexible event definition with a varying event duration, and a fixed definition and corresponding event duration. Both methods lead to a different number of events. This is relevant, as the impact of a set of loss events on an insurance is often not only dependent on the overall loss, but also on the number of individual loss events. Reasons for this are the Solvency II European Directive as well as the structure of reinsurance contracts (Vitolo et al., 2009), which take into account the frequency of losses and therefore also the number of events.

We furthermore aim to help close the gap between the multi-hazard research domain and meteorological clustering research by using impact data on multiple hazard types (i. e., multivariate extreme events) combined with observation data to categorize the events according to their meteorological drivers: pluvial, mixed and fluvial floods, hail, convective gusts, and large-scale storms.

The objective of this study is to address the following research questions:

1. When do clusters of individual damaging hazard types as well as clusters of their serial combination occur in southwestern Germany, and is this clustering significant compared to a random process?

2. How does the degree and significance of clustering depend on the chosen duration of the event?

3. Does clustering exacerbate the impact of hydro-meteorological hazards, as measured by insured losses?

This article is structured as follows: Section 2 provides an overview of the datasets used to identify distinct events and

subdivide them into meteorological categories, on the methods used to identify events, as well as on the methods used to identify and quantitatively assess temporal clusters. Section 3 describes how we have refined the hazard types into meteorological categories. In Sect. 4, we give an overview of the resulting loss distribution and seasonality of the events depending on their hazard type. In Sect. 5, we first explain how we combined events of different hazard types, then show and interpret the results of different metrics of clustering, a) a counting method and b) Ripley's K, before we discuss loss patterns and trends regarding

clusters. Section 6 elaborates on the main conclusions.





## 2 Data and Methods

This study is based on loss data (Sect. 2.1.1) from a building insurance company in the federal state of Baden-Württemberg
(BW) in southwestern Germany and covers the period from 1986 to 2023. All data were adjusted for inflation and the number
of contracts (Sect. 2.1.3), which vary substantially during the study period. Events lasting one or several days are identified
from the loss data using two different methods (Sect. 2.2). Meteorological observations (Sect. 2.1.2) were used to assign a
meteorological category to the loss data (Sect. 3).

### 2.1 Data

The region of BW has a size of approximately 36 000 km$^2$) and is characterized by the broad Rhine valley to the west and
the Neckar valley in the center, as well as the low mountain ranges of the Black Forest and the Swabian Jura. It represents
Germany's major hail hotspot (Puskeiler et al., 2016; Kunz et al., 2020). Heavy rain, which is particularly orographically
enhanced over the Black Forest, often triggers flooding in small to medium sized catchments (Kunz et al., 2023). Winter
storms may also be impactful, but are less frequent than in northern Germany, since BW is further away from storm tracks that
usually originate in the North Atlantic and propagate mainly towards northwestern Europe (Dacre and Pinto, 2020).

#### 2.1.1 Loss data

Extreme events, i.e. hydro-meteorological events leading to a major loss, are identified using data from a building insurance
company. The dataset includes residential building losses (with deductible subtracted) and the number of claims. The data
have a daily temporal resolution and are originally divided into storm, hail, and flood hazard types. During the study period,
the portfolio has expanded through the merger with direct insurance companies from other federal states. To determine the
overall loss for BW exclusively, we correct the dataset considering only the fraction of contracts for BW. Although this means
that the dataset may also contain events that did not affect BW onlyf, this step is necessary to ensure comparability over the
years. In addition, most of the contracts (around 85 % on average during the period under study) are from BW, so the resulting
uncertainty is relatively low. As there is no finer spatial information (such as e.g. on the municipality level) available in our
dataset, we use the spatially aggregated losses per day for the whole region. Due to the limited size of BW, we can assume that
there is only a single synoptic cause of major events at the same time.

In Germany, about 95 % of all buildings are insured against storm and hail damage (GDV, 2023). Other hazards, including
floods, can be insured with an additional insurance, the so-called elementary insurance. The coverage of this additional in-
surance is very heterogeneous in Germany, with a mean coverage of 52 % and an increasing overall trend over recent years,
ranging from 31 % of insured buildings in Bremen to 94 % in BW (GDV, 2023). The particularly high insurance coverage
in BW is mainly due to the fact that insurance was compulsory until 1994. Currently, about 60 % of all private buildings in
BW are covered by the data-providing insurer. Given the high settlement density in BW (Rösch and Treffinger, 2019), we can
assume that almost all events leading to significant damage are reflected in the data. We therefore did not further consider any
exposure correction of the data.



### 2.1.2 Meteorological data

The individual definitions of the three original hazard types in the loss data (flood, storm and hail) are not unambiguous. For
example, the storm category includes both winter storms and convective gusts, which are more likely to occur in summer.
Given the the different environmental conditions triggering these events, which also lead to different temporal and spatial
scales of the respective events, we further separate the storm and flood categories according to their main characteristics. For
this subdivision (see Sect. 3), we use meteorological observations from the German Weather Service (Deutscher Wetterdienst,
DWD).

For the flood hazard types, the *Hydrometeorologischer Rasterdatensatz Niederschlag für Deutschland* (HYRAS-DE-PRE)
dataset is used. It consists of station-based regionalized daily precipitation totals for Germany interpolated to the almost equidis-
tant $1 \times 1\,\mathrm{km}^2$ grid of the *Regionalisierung der Niederschlagshöhen* (REGNIE) product (Rauthe et al., 2013).

For the storm hazard types, the subdivision is performed using hourly measurements of surface pressure reduced to sea level
(mean sea level pressure, MSLP) at selected stations from the Climate Data Center (CDC) of the DWD (https://www.dwd.de/
EN/climate_environment/cdc/cdc_node_en.html). Data from six different weather stations distributed across and around BW
are used.

### 2.1.3 Data adjustment

The loss data are adjusted for both inflation as well as the number of contracts in the portfolio of the insurer, which varies signif-
icantly from year to year. Inflation adjustment is performed to the base year 2022 with the so-called *Gleitender Neuwertfaktor*
(glN), a factor commonly used in the German insurance industry (Dietz et al., 2015). This factor captures the development of
construction prices as well as standard wages and is published annually by the German Insurance Association. The inflation
correction factor is therefore defined as:

$$corr_{\mathrm{infl}}(year_x) = \frac{glN_{2022}}{glN_{\mathrm{year}_x}}. \tag{1}$$

Insured daily losses are then multiplied by this factor for the respective year:

$$loss_{\mathrm{adj}}(day_y) = loss(day_y) * corr_{\mathrm{infl}}(year_x). \tag{2}$$

The number of contracts is adjusted as follows: following the abolition of the insurance obligation in 1994 in BW, the
portfolio has declined almost continuously. We therefore additionally adjust the insured losses with a factor that captures the
number of contracts in the course of time, where $contractno_{\mathrm{mean}}$ refers to the mean contract number over the entire time
period:

$$corr_{\mathrm{contr}}(year_x) = \frac{contractno_{\mathrm{mean}}}{contractno_{\mathrm{year}_x}}. \tag{3}$$

The number of claims was adjusted by this ratio:

$$claims_{\mathrm{adj}}(day_y) = claims(day_y) * corr_{\mathrm{contr}}(year_x), \tag{4}$$



as well as the losses incurred:

$$loss_{\mathrm{adj}}(day_y) = loss(day_y) * corr_{\mathrm{contr}}(year_x).$$  (5)

These adjustments ensure a comparability across the time series, so that the loss data represent a solid basis for the assessment of clustering.

## 2.2 Identification of major loss events

Since our intention is to analyse the temporal variability and clustering of major loss events only, we retain data above the 90th percentile (p90) based on the daily loss data. The percentile filter is applied to both the damage claims and the insured losses

of the entire time series; values of zero, which relate to a loss lower than the deductible, are excluded before filtering. By using p90, we ensure to only capture relevant meteorological hazards. Note that this percentile filtering scheme leads to a different number of events for each hazard type.

Furthermore, as a prerequisite for applying extreme value statistics, the events are required to be independent. Toward this end, and to avoid clustering on the timescale of synoptic systems (around 5 days), clustering on the timescale of a few days

needs to be removed (Wilks, 2006). This is called (runs) declustering (Coles, 2001) and means, in our case, that the daily data are aggregated to events with a length of either one or several days.

Two different methods are used to define events: a) the Peaks-over-Threshold (POT) method, a standard method of extreme value theory (Sect. 2.2.1), and b) the Hours Clause (HC) method (Sect. 2.2.2), a method commonly used in the insurance industry (Mitchell-Wallace et al., 2017). We are not aware of any other study using the HC method or a comparison between

POT and HC methods for insurance loss data.

### 2.2.1 The Peaks-over-Threshold (POT) method

The POT method is widely used in hydrological and meteorological research (e.g., Barton et al., 2016; Tuel and Martius, 2021a) to model extreme events. We apply it as follows: first, we filter the loss data to retain only the most damaging events above a threshold $u$, which in our case corresponds to p90 (see Fig. 1a). We then apply runs declustering and aggregate events

above $u$ separated by less than $r$ days (or, more generally, time steps), i. e., count them as a single event (see Fig. 1b). A value of $r = 2$ is common for Central Europe regarding precipitation (Tuel and Martius, 2021a; Barton et al., 2016) and wind (Brabson and Palutikof, 2000) and therefore used here as well. For more details on the method, we refer to Barton et al. (2016). Resulting extreme events are mapped to a binary variable (i. e., we discard their actual associated losses, see Fig. 1), with a defined start and end date for each event, which can then be used as an input for the clustering analyses.

### 2.2.2 The Hours Clause (HC) method


The HC method is a commonly used method in the (re-)insurance industry to identify individual loss events. It relies on a pre-determined fixed event duration to obtain independent events. This event duration depends on the insurer as well as on the hazard type. We use an event duration of 72 h (=3 days) for storm and hail events and 168 h (=7 days) for flood events. These



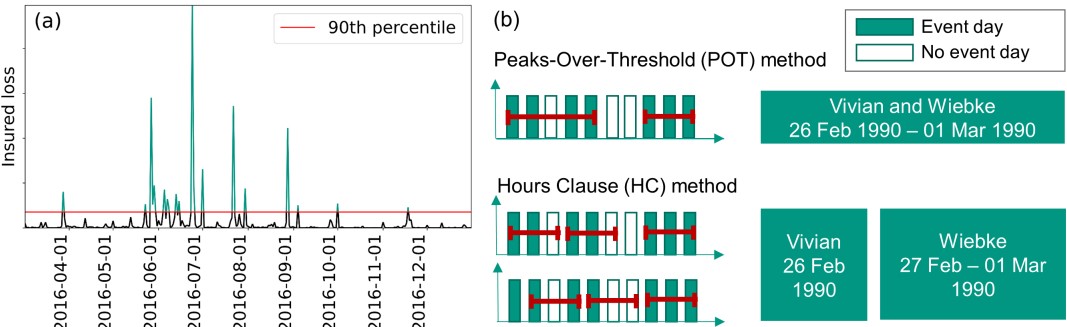

**Figure 1.** (a) Example time series demonstrating the Peaks-over-Threshold (POT) method, where only events above the 90th percentile are selected and (b) visualization of the two aggregation methods (POT; Hours clause, HC): event days and non-event days are aggregated to multi-day events; a winter storm example is consequently defined as one (POT) or two (HC) events.

values are also used by the building insurance company providing the data. We therefore calculate running sums of losses and claims over 3 and 7 days, respectively, depending on the hazard type, and aggregate over this non-filtered time series. For periods with loss records lasting longer than the predefined number of days, the day with the maximum loss is determined as the center of an event (day 2 for storm and hail and day 4 for flood events). Additional potential data points become further events assigned around this event (ending at its first day-1 day or starting at its last day+1 day). Then, the pre-defined events based on moving sums are filtered using p90. From this method, we obtain a sample of major damaging events for each type with a fixed length.

The HC and POT methods in some cases identify a different number of events. This can be seen, for example, for the severe winter storms Vivian and Wiebke in 1990, which are combined to a single event with the POT method, but identified as two separate events with the HC method (see Fig. 1b). In Sect. 5, the degree of clustering in the data is evaluated and compared for both methods.

## 2.3 Clustering methods

We combine two methods to assess serial clustering of the events, namely a counting method (Sect. 2.3.1) and the metric Ripley's K (Sect. 2.3.2). The former is used to identify time periods with an accumulation of extremes of single and multiple hazard types, while the latter is used to assess the degree of clustering by quantifying the deviation from a homogeneous Poisson process. Since the spatial extent of the study area is rather small, we exclusively investigate clustering in the temporal dimension.





### 2.3.1 Counting method

To count and thereby identify clusters, we adapt a method developed by Kopp et al. (2021). We first compute the number of (filtered) events $n_\mathrm{w}(t)$ for each hazard type and their respective combinations as well as their corresponding accumulated insured loss $il_\mathrm{w}(t)$ within forward-looking windows of a fixed length $w$ for each day $t$ of the time series $Y_t$ ($t = 1, \ldots, N$).

Building on this, we apply an algorithm that identifies cluster periods as follows. First, all counted time periods across the dataset with $n_w(t_0) = \max\limits_{t \in Y_t} n_w(t)$ are identified. Within that subset, the counted time period with maximum $il_\mathrm{w}$ is identified, which is the first cluster period. For that cluster period, the cluster start ($t_0$) represents the day on which the first event within the cluster starts. The end of the cluster period is defined by $t_0 + w$, independent of how many further events occur during this interval. Finally, to avoid overlapping clusters, all potential further events within $t_0 + w$ and $t_0 - w$ are removed. If there

are further periods for which $n_w(t_0) = \max\limits_{t \in Y_t} n_w(t)$, the second cluster is identified with the second highest $il_\mathrm{w}$, and so forth. If no clusters of $\max\limits_{t \in Y_t} n_w(t)$ remain, the subsequent cluster is identified at $\max\limits_{t \in Y_t} n_w(t) - 1$. This procedure is continued until no further clusters can be identified, i. e., $\max\limits_{t \in Y_t} n_w(t) < 2$.

### 2.3.2 Ripley's K

To quantify the degree of clustering, we employ the statistical tool Ripley's K (Ripley, 1981), which is a function originally

applied to quantify the clustering of point patterns at varying spatial scales. Ripley's K has also been applied to one-dimensional time series of meteorological or hydrological extremes (e.g., Barton et al., 2016; Tuel and Martius, 2021a, b; Brunner and Stahl, 2023). For a time series and clustering window $w$, Ripley's K is defined as the average number of events $E(w)$ within a time window $w$ (here in days) around any event in the time series:

$$K(w) = E(w), \tag{6}$$

and can be estimated, e. g. by:

$$\hat{K}(w) = \left[ \frac{1}{N} \sum_{n=1}^{N} \sum_{k=-w}^{w} Y_{t+k} | Y_t = 1 \right] - 1, \tag{7}$$

as in e. g. Barton et al. (2016), where $Y_t$ relates to the binary time series. Ripley's K therefore quantifies the average number of events (major loss events in our case) within $\pm w$ days of an event. We let the time window $w$ range from the timescale of a few days (note that due to the identification of independent events, $K = 0$ for the first few days) up to the seasonal level ($w =$

45 or $w = 60$, depending on the length of the season). Clustering on the seasonal level in our case compares the occurrence of events between different years (=seasons).

As in Barton et al. (2016) and Tuel and Martius (2021a), the statistical significance of the clustering is tested by a comparison with a random homogeneous Poisson process, which consists of independent events and is therefore characterized by temporal randomness. It is simulated here by 1 000 Monte Carlo runs with the same probability of occurrence (or: average density of

events) than the observed extremes. These simulations are also declustered, using POT for both methods of event identification. Due to the strong seasonality present in the data (see Sect. 4.2), each month is simulated separately. For each $w$ where the



observed $K(w)$ exceeds the 95th percentile of the simulated $K(w)$, the data are significantly clustered. Conversely, where the observed $K(w)$ is lower than the 5th percentile of the simulated $K(w)$, the data are significantly regularly spaced (Barton et al., 2016). Otherwise, the series cannot be statistically distinguished from a homogeneous Poisson process. For a detailed
explanation of the significance analysis and p-values, see Tuel and Martius (2021a).

Since the Ripley's K function can only take one event date per event as input, we use the start of each event (identified with POT or HC) as an input timeseries to Ripley's K. We take only hazard types with > 20 events throughout the time series into consideration. To avoid artificial clustering caused by the recurring seasonal patterns (see Sect. 4.2), we analyze the data for specific seasons separately: May to August (MJJA) and December to February (DJF).

## 3 Categorization into meteorological hazard types

Before clustering, we further subdivide the storm and flood loss events in our loss dataset. In fact, both types of events can be attributed to different underlying mechanisms. Convection-driven events usually extend over a few to a few dozen kilometers only, and last less than one hour. By contrast, events that are triggered and maintained by large-scale lifting processes in the mid-troposphere usually extend across several hundreds to thousands of kilometers, and may persist from several hours to
several days. These two types of events will therefore affect substantially different areas, a fact which must be considered in the event definition and for the cluster analysis.

### 3.1 Flood damage events

A predominantly stratiform precipitation event is characterized by low to moderate rainfall intensities, a duration of several hours to days, and a large affected area, sometimes extending over several hundred kilometers (Houze, 1993). It can result in
fluvial floods, i.e., rivers breaking their banks. By contrast, a convection-dominated precipitation event is characterized by high rainfall intensities combined with a short duration of a few minutes to a few hours, and has a small spatial footprint. It can result in pluvial floods. However, a clear separation between these two hazard types is not always possible, in particular for mesoscale convective systems (MCS), in which convective activity is embedded within a stratiform precipitation field (Cannon et al., 2012); for clustered convective events with a mixture of stratiform and convective precipitation primarily toward the end of the
life cycle (Houze, 1993); or for orographic precipitation. The latter can attain very high precipitation totals, particularly over the low mountain ranges of the Black Forest (Kunz, 2011). Therefore, we define three categories of predominantly stratiform, predominantly convective and mixed precipitation, typically leading to fluvial, pluvial and mixed floods respectively.

The most straightforward method to distinguish between these hazard types would rely on radar data (e.g., Wang et al., 2021). However, radar data are not available for the entire period of our study. Therefore, we use gridded daily precipitation
totals from HYRAS-DE-PRE during identified insurance loss events. Two metrics are considered for the separation between the precipitation types: the 99th percentile, and the coefficient of variation (CV) of daily gridded precipitation totals (calculated in space). The CV, also called normalized standard deviation, is defined as the ratio between standard deviation $\sigma$ and mean $\mu$





(Abdi, 2010):

$$CV(t) = \frac{\sigma}{\mu}.$$

For $CV(t)$, only grid points with precipitation (i. e., > 0 mm) are considered.

In order for a day t to be classified as convection-dominated (associated with pluvial flooding), the following conditions must be met: (i) the spatial 99th percentile of daily precipitation totals exceeds 40 mm for any day from t-2 to t, (ii) $CV(t)$ is larger than or equal to 0.55 mm, and (iii) t is between 1 April and 30 September (since convective activity primarily occurs during the summer months in Germany (Kunz, 2007; Mohr et al., 2017)). We go back up to t-2 because of a potential time lag

between the precipitation and the flood damage, particularly in case of a stratiform-dominated event or for heavy rain around midnight. Remaining days are categorized as mixed floods if $CV(t) > 0.45$ mm, and otherwise as stratiform-dominated (fluvial flood) events if the mean precipitation for all grid points is larger than 2 mm for that day. These thresholds were selected and extensively tested by comparing to radar images from DWD (available after 2005).

Of course, the above described criteria do not separate all events clearly. Therefore, uncategorizable events – mainly those

around the two defined thresholds – were visually checked and reassigned with expert knowledge partly in combination with existing literature (e.g., Kunz, 2003) as well as by taking into account the terrain height (e.g., Brommundt and Bárdossy, 2007). In cases of multi-day events, the hazard type of the day with the largest precipitation totals in the (spatial) 99th percentile was assigned to the entire multi-day event. Hereinafter, we refer to the stratiform-dominated flood hazard as fluvial flood, to the mixed flood hazard as mixed flood and to the convection-dominated flood hazard as pluvial flood.

## 3.2   Storm damage events

Damaging hazard types related to the storm category are typically either so-called windstorms/extratropical cyclones that extend on a synoptic scale (around 1 000 km), or (severe) convective gusts that have much smaller spatial scales (tens of km) (Markowski and Richardson, 2011). As with precipitation, insured losses categorized as storm damage can also result from the interaction of local- and large-scale processes. During the passage of cold fronts of winter storms, for example, convection

can strengthen surface winds by convection-driven downbursts (Markowski and Richardson, 2011; Mohr et al., 2017). In case of convective gusts, the vertical transport of horizontal momentum increases convective gusts at the surface. However, all these processes occur on a local scale and cannot be simply estimated from available observation data. In addition, our main purpose here is to separate between damage created by the different triggering mechanisms, which presumably feature different clustering characteristics. For these two reasons, we only distinguish between synoptic- and convection-driven storms and do

not consider a mixed class for storm events as we do for the flood hazard.

To differentiate between these two main storm hazard types, we use the method of Mohr et al. (2017) using the MSLP gradient between selected weather stations. We consider hourly MSLP observations from six DWD weather stations available from the CDC for the entire investigation period. For three axes across BW, station pairs were created: Northwest-Northeast (stations Karlsruhe/ Rheinstetten and Weißenburg-Emetzheim), Southwest-Southeast (stations Freiburg and Lechfeld) and North-South

(stations Michelstadt-Vielbrunn and Konstanz). For each of these pairs, we compute the MSLP gradient as the ratio of the



difference in MSLP to the distance between the stations. If, between any station pair at any time on a specific day, this MSLP gradient exceeds a threshold of $3\,\mathrm{Pa\,km^{-1}}$, the day is defined as a synoptic storm day. Otherwise, it is classified as a convective storm day. The threshold was slightly reduced compared to Mohr et al. (2017), after testing with severe winter storms in recent years. For multi-day events, if any day is classified as synoptic, we classify the entire multi-day event as synoptic. In the

following, we refer to storms with a synoptic trigger as large-scale storms and to convection-driven storms as convective gusts.

## 4 Loss distribution analysis

Before presenting the results of the clustering methods, this section gives an overview of the loss dataset we use, in particular the distribution of losses and their seasonality. If not stated otherwise, events are identified using the POT method for these analyses.

### 4.1 General loss distribution

Regarding cumulative insured losses and damage claims, two large-scale storms and two hail events (Fig. 2a) were responsible for 30 % of the total cumulative insured losses in the entire period (18 % of all claims). More generally, only 3 % of events are responsible for 86 % of the insured losses throughout the time series. The Gini coefficient, which is a statistical measure for inequality in a distribution, equals 0.96 (1 relates to perfect inequality) when comparing insured losses throughout the dataset.

This shows that the loss distribution among the dataset is strongly right-skewed. Similarly skewed distributions are found for Europe between 1980–2022, where 1 % of all climate-related events account for 28 % of insured losses (European Environment Agency, 2023).

When removing major events (Fig. 2b), we find that the amount of insured losses is strongly related to the number of damage claims per hazard type. Indeed, insured losses show a high correlation with the number of claims, especially for large-scale

storms (Pearson's r = 0.95), hail and mixed floods (r = 0.94) as well as fluvial floods (r = 0.93), and their relationship can be described by a linear function for each hazard type. This has important implications for insurance loss modeling, notably to set risk premiums.

These relationships imply that the mean loss per claim for each hazard type does not vary significantly with the extent of an event, even in case of events that affect a large area (i. e., a high number of claims). This implies that the number of damage

claims could be used as a proxy to estimate total damage in the absence of loss data. Second, mean loss patterns, i. e., loss per damage claim, differ substantially depending on the hazard type. Fluvial floods cause the highest mean losses of all hazard types, while large-scale storms cause the lowest mean losses (without major events).

An explanation for the differences in mean losses could be the nature of the damaging hazard type: Large-scale storms, by definition, affect a large area, much of it only suffering from low damage, e. g., some removed roof tiles. By contrast, hail

events occur locally and have a higher damage potential, since more parts of a building become susceptible (Stucki and Egli, 2007). Hail damage can result from e.ġ. the piercing of roof tiles, roof windows, solar installations, facades, shutters, or winter gardens. In case of a damaged roof, additional damage can result from a subsequent rain event (even of moderate intensity),





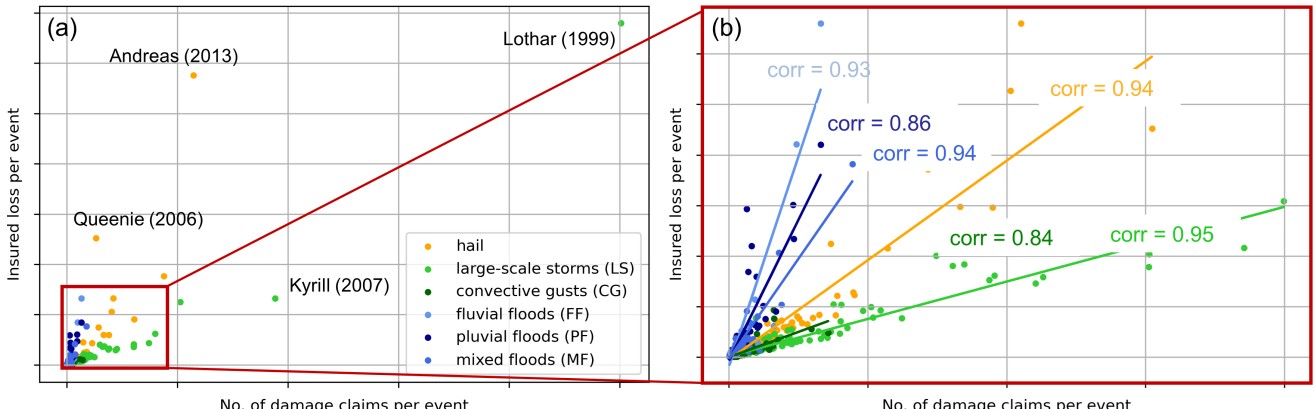

**Figure 2.** (a) Total insured losses versus number of claims for building insurance for hail damage, damage by convective gusts and large-scale storms, and pluvial, fluvial and mixed flood damage, in BW from 1986 to 2023, events defined and aggregated using p90 and POT. Four of the most damaging events in terms of losses and/or claims are indicated. (b) As in (a), but without major damaging events, including linear fit and Pearson correlation coefficients for the respective hazard type.

where water can enter the building through the roof (Stucki and Egli, 2007). The fact that flood hazard types show the highest mean loss is especially relevant considering that there is no mandatory insurance against floods for residential buildings (GDV, 2023). This highest mean loss might be due to a flood event being more likely, compared to hail or storm events, to affect the interior of a house as well as the exterior. Entire floors can get flooded once the water has entered (Merz et al., 2010), and reconstruction is a tedious and expensive undertaking (e.g., Mohr et al., 2023).

Within flood hazards, although fluvial events are related to the highest mean losses, pluvial events lead to the highest annual insured losses of all flood hazards (not shown). This might be due to the absence of an extreme fluvial flood event in BW during the period considered. Since other regions in Germany have experienced severe floods in the last decades (e. g., Elbe in 2002 or Ahr and other rivers in western Germany in 2021), BW appears to be an exception in this regard. Mixed floods, even though they are the most frequent flood hazard throughout the year, are less damaging in terms of mean annual losses. This lower impact of mixed events suggests that the more significant damage is driven by distinct meteorological systems – either stratiform or convection-dominated – rather than by mixed events. When major events are eliminated from the sample, the highest mean losses within the flood hazard are caused by pluvial floods. Regarding storm damage events, convective gusts are most frequent, but mean and cumulative losses are highest for large-scale storms. Again, when removing major events, convective gusts lead to the highest mean losses. This shows that, in addition to the hail hazard, a relatively small number of major large-scale flood and storm events can cause extreme damage in Germany. The convective hazards, namely pluvial floods and convective gusts, are of secondary importance in our data set in terms of the damage they cause.




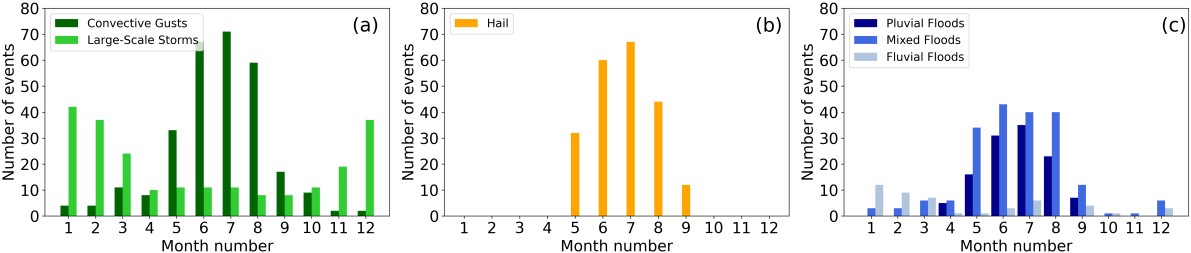

**Figure 3.** (a) Distribution of events identified with POT in p90 over aggregated months of flood damage triggered by pluvial, fluvial and mixed floods, convective gusts, large-scale storms and hail damage events in BW from 1986 to 2023.

## 4.2 Seasonality

All major damaging hazard types show a strong seasonality, with most events occurring in summer (MJJA) for most hazard types (Fig. 3). Hail damage in the winter half year is marginal compared to the summer months. Therefore, hail damage occurring between October and April was reassigned to the storm loss category prior to event identification with POT or HC.

All convection-driven hazard types, i. e., hail, convective gusts, and pluvial floods, show a similar seasonality: The number of events peaks in June or July, with significantly fewer events in August and even fewer in May, when the convective season starts (Taszarek et al., 2020). In August and September, convective storms occur more frequently compared to hail, which is robust with regard to the event loss threshold (p95), but with a weaker pattern (not shown). This might indicate that the damage-related convective gusts are less likely to be accompanied by (damaging) hail in BW in late summer. Mixed floods are the most common flood hazard causing extreme damage from April to September. In contrast to the solely convection-driven events, there is a similar number of mixed flood events throughout MJJA without a strong fluctuation.

Both large-scale storms and fluvial floods occur mainly during the winter months. Damaging windstorms show a peak in DJF, which follows the general seasonal distribution of extreme wind speed (Gliksman et al., 2023). Most fluvial floods occur between January and March. Thus, we see a strong seasonal pattern of the occurrence all hazard types, with a smaller number of large-scale events being dominant in the winter months and a higher amount of local extremes more relevant in summer.

## 5 Clustering

When analysing temporal clustering on the timescale of calendar years with POT (Fig. 4), we find that the number of damaging events varies substantially over time. There is a pronounced peak in the early 2000s for both storm hazards, a peak between 2017 and 2019 for hail events, a smaller number of storm and hail events before 1998, and an especially high inter-annual variability of the flood hazard types. The HC event definition generally identifies a smaller number of events than POT, although the general distribution of events throughout the years is similar. For both definitions, we see a kind of wave pattern throughout



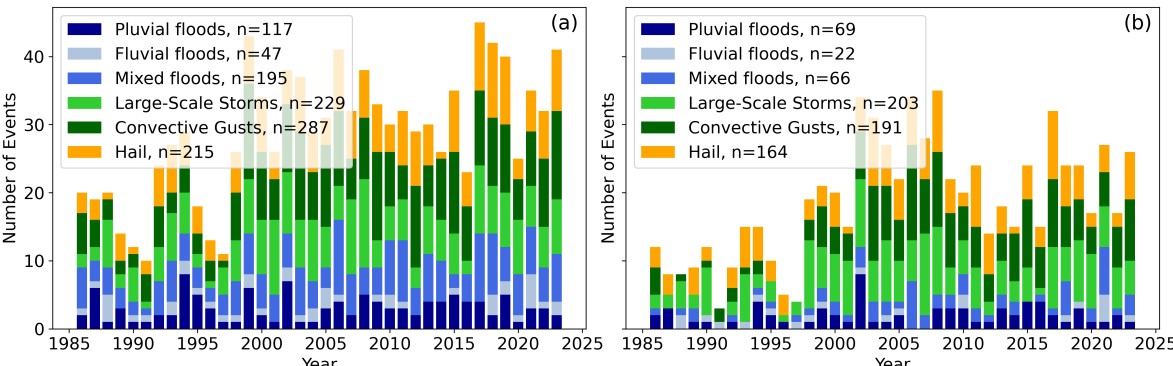

**Figure 4.** Number of events per hazard type and year with (a) event definition with POT, (b) event definition with HC, for events defined by the 90th percentile (see Sect. 2.2).

the years, with some years showing an exceptionally small number of events. This wave pattern could be related to decadal variability, which has been discussed with regard to hail in Europe by, e. g., Mohr et al. (2015).

## 5.1 Combination of hazard types

Before analyzing clustering of single hazard types and their combinations, we here explain how the different hazard types are
combined. Single hazard types include pluvial floods (PF), mixed floods (MF), fluvial floods (FF), large-scale storms (LS), convective gusts (CG) and hail (H). We refer to the combinations between events of different hazard types by acronyms that combine those of the single hazard types. Therefore, a combination of pluvial floods, convective gusts and hail, for example, is referred to as PF-CG-H. All combinations between the three flood hazards and the two storm hazards as well as hail are performed.

When events of different hazard types are combined, they again need to be declustered to avoid artificial clustering of events from the same meteorological driver (see Sect. 2.2 for declustering of single hazard types). We perform the following procedure: if a day experiences several hazard types, it gets assigned to the hazard type with the highest insured loss and is counted as one event. However, all hazard types are retained, so they can potentially lead to clustering of multiple hazard types (e. g., if a hail event and convective gust event occur on the same day and another convective gust event occurs within the
clustering window, this is defined as a combined cluster of hail and convective gusts only because both types are retained).

## 5.2 Cluster identification: Counting

Figure 5 shows the resulting cluster periods for (a) single and (b) combined extremes for clustering window $w = 21$ and events identified with POT. It is evident that most clusters occur during MJJA, consistent with the seasonal distribution of the events (see also Fig. 3). These are mainly clusters of multiple hazard types, with the most damaging combination being PF-CG-H,
followed by PF-H. This means that the most damaging clusters consist of three hazard types which are registered separately




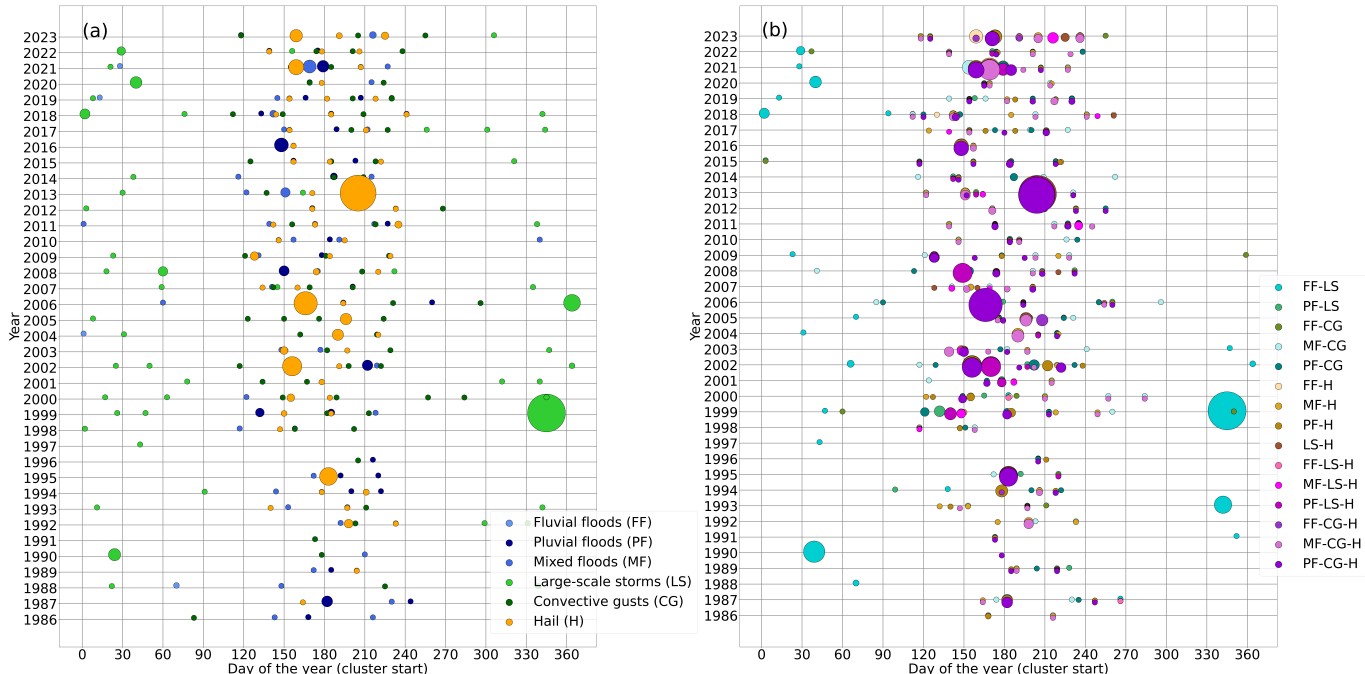

**Figure 5.** Cluster start by the day of the year (x-axis) plotted against the corresponding years (y-axis), identified by the counting method with a clustering window w of 21 days for (a) single and (b) combined hazard types (p90, POT method). Size of the circles relates to the loss corresponding to each cluster (normalized).

by the insurance company. In winter (DJF), we see a much lower number of clustered events. Predominant in this season are clusters of large-scale storms and FF-LS, which also aligns with the seasonal cycle of these hazard types.

Throughout the years, we see a peak of clusters during May–August of the early 2000s, which resembles the general pattern of event occurrence. Interestingly, this is not the case for clusters in the winter months. Winter clusters were most frequent, and also most damaging, in the 1990s as well as since 2017. Between 2005 and 2018, there are very few clusters of different hazard types in winter, which does not relate to a generally lower event number (cf. Fig. 4). The number of clusters in summer is also quite low in some of those years, e. g. in 2014 and 2015. Seasonal patterns are visible even more clearly with clusters of different hazard types (Fig. 4b) than with single-hazard type clusters (Fig. 4a). There are almost no clusters between mid-February and mid-April, and similarly very few clusters between the end of September and the end of November. Note, however, that this refers to the cluster start date (e. g., clusters starting in February might contain events in March). Furthermore, it can be seen that many of the single-type clusters also occur in multi-hazard clusters, particularly the most damaging ones. This shows that many clusters consist of multiple hazard types, but also include multiple occurrences of a single hazard type. Note that the number of events is much higher when assessing clustering with multiple hazard types, i.e., naturally, a higher number of clusters can be found.



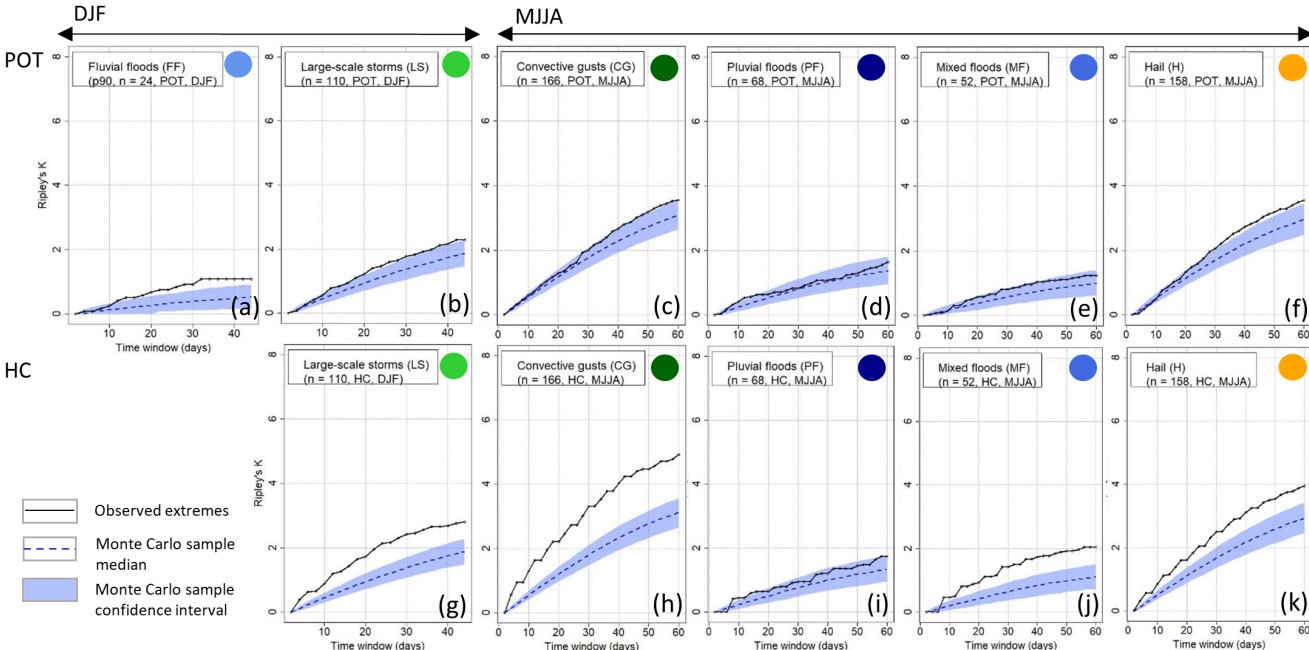

**Figure 6.** Clustering results using Ripley's K for single hazard types: large-scale storms, pluvial floods, convective gusts, fluvial floods, mixed floods and hail damage, depending on the season (columns) and method of events identification (rows).

When events are identified using the HC method, the number of clusters is reduced (not shown), since the number of events is also lower. However, this mainly relates to the least damaging clusters; the most severe clusters are similar to those detected with the POT method. Using the HC method, some additional clusters are detected, such as a mixed flood cluster in 2021 and several multi-hazard clusters in 2001. Hail clusters are most frequent as well as those clusters leading to the highest cumulative losses when events are identified with HC.

These results are tested for robustness by increasing the clustering window $w$. For $w = 28$ days, for example, the number of clusters is higher in summer, and slightly shifted to earlier start dates in winter.

### 5.3 Clustering assessment: Ripley's K

The number of events influences the degree of clustering in a time series (recall the differences in event numbers between POT and HC, see Fig. 4). Therefore, when applying Ripley's K, the number of events identified by POT for each hazard type are reduced, using a ranking of insured losses, until the number of events identified with HC for that hazard type is reached.



### 5.3.1 Clustering of single hazard types

Due to the seasonality of the events (see Sect. 4.2), we investigate large-scale storms and fluvial floods in winter (DJF) and convective gusts, pluvial and mixed floods as well as hail in summer (MJJA). Generally, Ripley's K for events identified with the HC method ($K_{HC}$) is significant on a broad range of timescales compared to a random series. It is also higher than Ripley's

K for events identified by the POT method ($K_{POT}$; Fig. 6).

Overall, the strongest clustering, which is significant against the 95 % confidence interval of a Monte Carlo sample (hereinafter: significant), is found for convective gusts during MJJA with $K_{HC}$: at a seasonal timescale, around five additional convective gust events can be expected around a random convective gust event. However, for $K_{POT}$, i.e., a flexible event definition, the time series of convective gusts is within the 95 % confidence interval on almost all timescales and only slightly exceeds it

around the seasonal scale. We see a similar pattern for mixed floods during MJJA: $K_{HC}$ is significant on all timescales, whereas $K_{POT}$ is not significant. Pluvial floods do not cluster significantly during MJJA with both methods of event identification. Hail clusters are significant on all timescales with $K_{HC}$, but only from the timescale of 30 days to seasonal level for $K_{POT}$. During DJF, fluvial floods cluster significantly on the timescale from 20 days to the seasonal level with $K_{POT}$, but with low values of K, i. e., a low number of surrounding events (due to a generally low number of events). This cannot be compared to $K_{HC}$ because

of a very small sample size (15 events). Large-scale storms during DJF cluster significantly on all timescales with $K_{HC}$, and are significant with $K_{POT}$ starting from about 20 days, but with a low difference from the 95% confidence interval.

To assess the robustness of the results, we systematically changed various variables. Changing the seasonal focus from MJJA to only JJA, $K_{POT}$ is not significant for convective gusts and hail. This implies that the occurrence in JJA follows a homogeneous Poisson process, but clusters in those months if May is added. Interestingly, when events are identified using p95

instead of p90, we see an increased degree of significant clustering for hail and convective gusts with $K_{POT}$, but a decreased degree of clustering for $K_{HC}$. For the flood hazards, there is only little change. If no declustering is applied, events do cluster significantly in both seasons and for all original hazard types as defined by the insurance company (storm, hail and flood). This shows that clustering occurs at short timescales, which is why declustering is needed. The results do not change when increasing the number of simulations for the significance test.

One reason for the increased number of significant results concerning clustering of events defined with the HC method is the duration of events. Due to the definition of HC, the duration of any event cannot be lower than three days (or seven days for flood events). However, the average duration for events identified with the POT method is 1.92 days. This on average much lower duration of events identified by POT compared to those identified by HC influences the degree of clustering. Although both methods can only approximately reproduce the actual duration of events, because of the only daily temporal resolution of

the underlying loss data, it should be noted that POT is clearly more accurate because of its flexible nature. We therefore argue that by using HC, the degree of clustering is often overestimated.

In their global analysis of precipitation extremes, Tuel and Martius (2021a) find low values of Ripley's K for European regions and detect significant clustering over Europe only for a few grid cells for both DJF and JJA. For heavy precipitation events in Switzerland, defined using POT from gridded daily precipitation data, Barton et al. (2016) show similar results. For





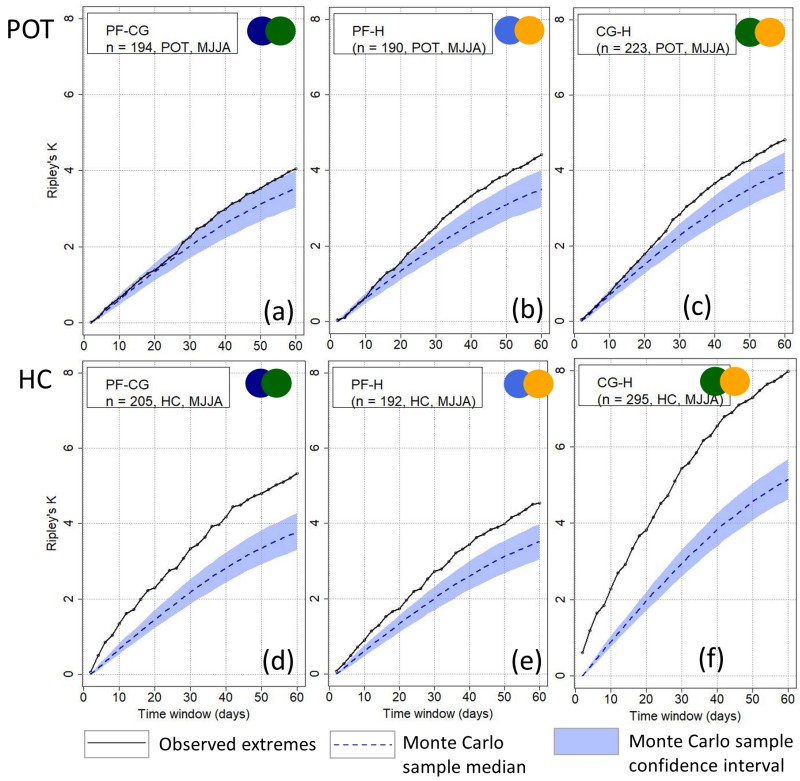

**Figure 7.** Clustering results using Ripley's K depending on the method of events identification (rows) and during MJJA for the combination of two hazard types including pluvial floods (PF), hail (H) and/or convective gusts (CG) (part 1).

DJF and JJA, they find no significant clustering on the seasonal timescale for p95 when declustering is applied. Tuel and Martius (2021b) similarly detect low values of Ripley's K and no significant clustering of heavy precipitation for most Swiss regions during DJF and JJA. This is in line with our results for southwestern Germany, since we detect low and not significant values of Ripley's K for all three flood hazards in both DJF and MJJA.

   To our knowledge, clustering of extratropical cyclones has not yet been assessed with Ripley's K, but is mainly investi-
gated using the dispersion statistic. A statistically significant overdispersion, indicating clustering, is identified specifically in northwestern Europe, over the exit region of the North Atlantic storm track (e.g., Mailier et al., 2006; Vitolo et al., 2009), summarized in Dacre and Pinto (2020). This region however does not clearly extend to Germany. Dacre and Pinto (2020) also highlight that in Europe, more intense extratropical cyclones tend to cluster more frequently than larger samples of cyclones including also less intense ones in Europe, as shown across multiple studies. We find contrasting results: when we decrease our
sample size towards more extreme large-scale storms, we find a decreased degree of clustering.

   For hail events, to our knowledge, there is no systematic assessment of temporal clustering on the seasonal scale.



### 5.3.2 Clustering of two hazard types

When we apply Ripley's K to a combination of two hazard types, no combination may consist of more than 80 % of a single hazard type throughout the (seasonally filtered) time series. This prevents a particular type of hazard from dominating the cluster. With this condition, the combinations pluvial floods-convective gusts (PF-CG), mixed floods-convective gusts (MF-CG), convective gusts-hail (CG-H), mixed floods-hail (MF-H), pluvial floods-hail (PF-H), and pluvial floods-large-scale storms (PF-LS) are feasible for MJJA. For DJF, the combinations large-scale storms-mixed floods (LS-MF) and large-scale storms-pluvial floods (LS-PF) are feasible.

Ripley's K results (Fig. 7 and 8) show that $K_{HC}$ is significant for all feasible event combinations during MJJA for all timescales from a few days up to a season, with the exception of PF-LS. The degree of clustering is highest for CG-H. On average, eight events are found around a random event in the time series at the seasonal scale, which significantly deviates from a homogeneous Poisson process. This is probably due to the strong degree of clustering of convective gusts (see Fig. 6), but also due to the strong clustering of hail. Convective gusts and hail often occur in close succession if an unstable air mass prevails for several days. The significance of $K_{POT}$ is more pronounced when two hazard types are combined compared to the single hazard types. $K_{POT}$ is significant for PF-H and CG-H from the timescale of about 20 days, but not for PF-CG (see Fig. 7). $K_{POT}$ for MF-CG and PF-LS respectively does not significantly differ from a homogeneous process. For the combination of MF-H and MF-LS (see Fig 8, $K_{POT}$ is significant from about 30 days.

In summary, we see significant clustering for combinations of two hazards in MJJA for $K_{HC}$. Concerning $K_{POT}$, the results suggest that combinations of two hazard types involving hail lead to clustering. The combinations cannot be quantitatively evaluated for DJF due to the low sample size. When decreasing the number of events up to the 95th percentile, the degree of clustering decreases, as with the single hazard types.

### 5.3.3 Clustering of three hazard types

For combinations of three hazard types, we introduce an additional condition: each hazard type must account for at least 10 % of the total event count (per season) of all three hazard types. Without this requirement, the combination could include a very small number of events from one hazard type, leading to clustering results that effectively reflect only events from the other two hazard types.

The combinations MF-CG-H, MF-LS-H, PF-CG-H and PF-LS-H fulfil this condition within MJJA. As with the combination of two events, $K_{HC}$ is higher and more often significant than $K_{POT}$ in almost all cases, especially where CG and H are involved (Fig. 9). For the POT method, we also find significant clustering for all combinations of three hazard types, at least at the seasonal scale and starting from 10–30 days. The occurrence of combinations of three damaging hazard types during MJJA therefore differs significantly from a homogeneous Poisson process at timescales of 30 days up to a season, regardless of the definition of events. When reducing the number of extremes to p95, the degree of clustering remains similar.

Overall, for events identified by POT, the clustering of the combination of several hazard types often starts from the timescale of about two to three weeks. For Germany, Bloomfield et al. (2023) show that correlations between storm and flood events are



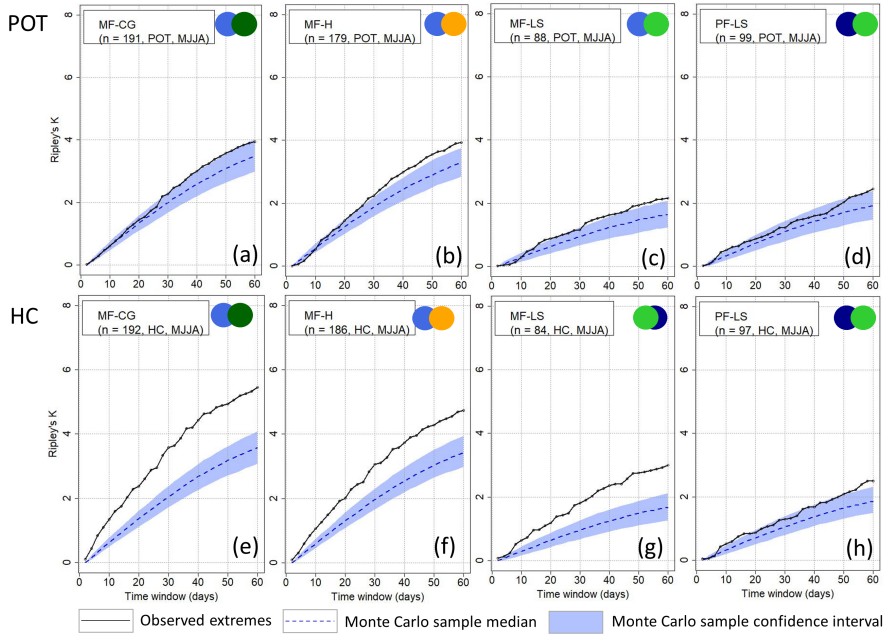

**Figure 8.** Clustering results using Ripley's K depending on the method of events identification (rows) and during MJJA for the combination of two hazard types including pluvial floods (PF), mixed floods (MF), large-scale storms (LS), hail (H) and/or convective gusts (CG) (part 2).

highest at a monthly scale (impacted by storm clustering). Therefore, with a counting window of 21 days (see Sect. 5.2) we should be able to detect most of the clusters. It can also be seen that for $K_{\text{POT}}$, where large-scale storms, convective gusts, pluvial floods and mixed floods do not cluster on most timescales during MJJA (Fig. 6), the combination with other hazard types increases their degree of clustering. This means that the approach of analysing single hazard types only could overlook a cluster due to the occurrence of other hazard types.

To our knowledge, there are no other studies quantifying the degree of temporal clustering with respect to different types of (meteorological) hazards. We therefore contribute to the literature by considering a variety of meteorological hazard types and finding that they do cluster, irrespective of the event definition.

## 5.4 Loss patterns of clustered events

For large-scale storms and fluvial floods during DJF, the median loss of clusters within 21-day windows (n=37) exceeds the
median loss of isolated events (n=91) by a factor of 4 (not shown). This pattern also holds for clustering windows of 14 or 28 days, highlighting that multi-hazard clusters lead to higher losses during DJF compared to isolated hazards. A similar result is found for overall Germany by Xoplaki et al. (2023), who show a much higher loss ratio for residential buildings regarding co-occurring wind and precipitation extremes in winter compared to their isolated occurrence. For the UK, Hillier et al. (2015)

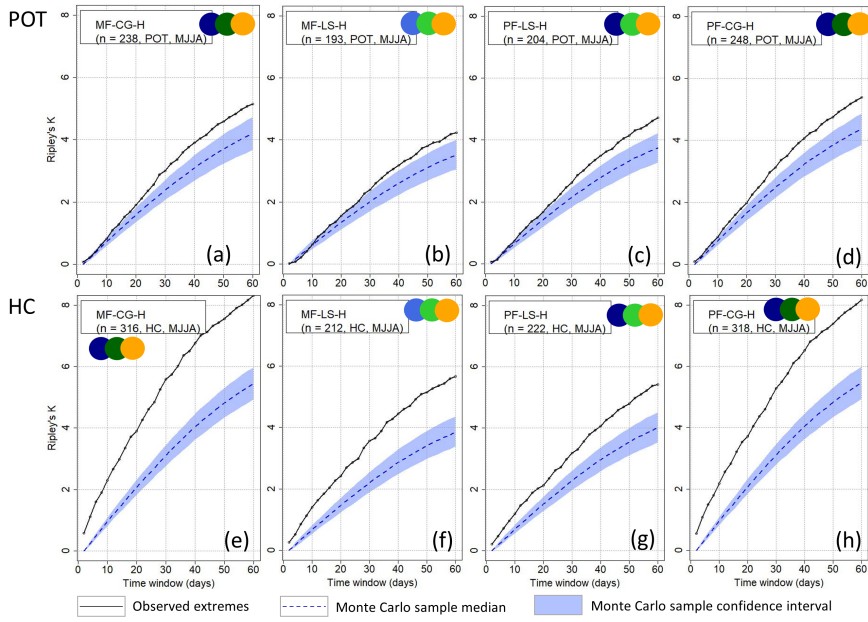

**Figure 9.** Clustering results using Ripley's K depending on the method of events identification (rows) and during MJJA for the combination of three hazard types including pluvial floods (PF), mixed floods (MF), hail (H), large-scale storms (LS) and/or convective gusts (CG).

show, based on rail data, that interactions between floods, winter storms and shrink-swell subsidence events increase insured losses by up to 26 % per year.

Figure 10 shows that this phenomenon is also present for convective clusters during MJJA: Clusters of PF-CG-H lead to higher losses (median loss increased by a factor of 1) compared to the isolated occurrence of any of these hazard types. Note that this specific hazard combination also leads to the highest degree of clustering. The most damaging clustered events include, for example, the hail event *Andreas* on 28 July 2013 (see Fig. 2), which was accompanied by pluvial flood damage and preceded by another pluvial flood as well as convective gusts on 23 July 2013. Another cluster with high losses includes the hail event Queenie on 28 June 2006, which was accompanied by a pluvial flood, and preceded by convective gusts and hail on 25 June.

The substantial amplification of losses by clusters of damaging events from different hazard types highlights the importance to consider this effect in applications such as risk modelling. This is even more important, as damaging hazards of different types frequently occur in close succession during persistent synoptic settings or weather patterns, such as blocking or an extended Atlantic trough (Grams et al., 2017), which can trigger individual extremes.

## 5.5 Trends

From 1986 to 2023, the number of damaging extremes in BW has increased significantly, and the same can be said for clusters of damaging extremes (see Fig. 11). Note that although the linear regression only explains a limited part of the variance due to the strong annual variability, the upward trend is clearly significant (p-value < 0.0001 for all events, p-value = 0.0002 for



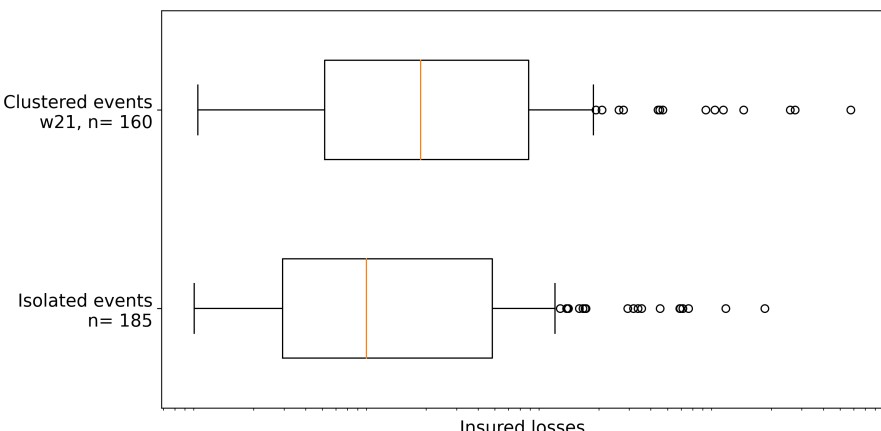

**Figure 10.** Insured losses (logarithmic) of clustered vs. isolated extremes within a clustering window of 21 days, here related to the combination of pluvial floods, convective gusts, and hail. Loss amounts are not shown due to their confidentiality.

clustered events). This upward trend is also significant with events that occur in clusters of 14 days. When investigating all multi-hazard clusters separately, there is a significant increase in clusters consisting of MF, CG, MF-CG, MF-H, and MF-CG-H. The number of clustered events of other hazard types or combinations does not increase significantly throughout the time frame. Note however that these upward trends are also governed by increasing values of the objects and vulnerable extensions, such as winter gardens or solar panels. Additionally, throughout the entire period, both the reporting and regulation of claims

have undergone substantial changes.

Since all of the natural hazards under consideration occur seasonally, the share of events within clusters compared to all events within a year is quite high (65 % with a clustering window of 14 days, 83 % with a window of 21 days, and 86 % with a window of 28 days). However, certain years stand out: In 1988, 1996, 1997 and 2016 less than half of the events occur within clusters of 14 days. This share of clustered extremes compared to all extremes has increased throughout 1986–2023 by about

8 %, even though this increase is statistically not significant.

The overall annual losses have also increased throughout the past years by about €1.5 million per year (adjusted for inflation, not significant either). Furthermore, this increasing trend is also influenced by non-meteorological factors, which could not be factored in. In the literature, for overall Germany, an increasing trend regarding storm and hail damage is found (GDV, 2023). Globally, there is an increasing trend of inflation-adjusted insured losses by about 3 % per year (Banerjee et al., 2024).

**6   Conclusions**

In this study, we have assessed the occurrence and degree of clustering of multiple meteorological hazard types (hail, pluvial, fluvial and mixed floods, convective gusts and windstorms) in southwestern Germany based on building insurance loss data. We have shown that random clustering of damaging meteorological hazard types and their combinations exists. Clustering



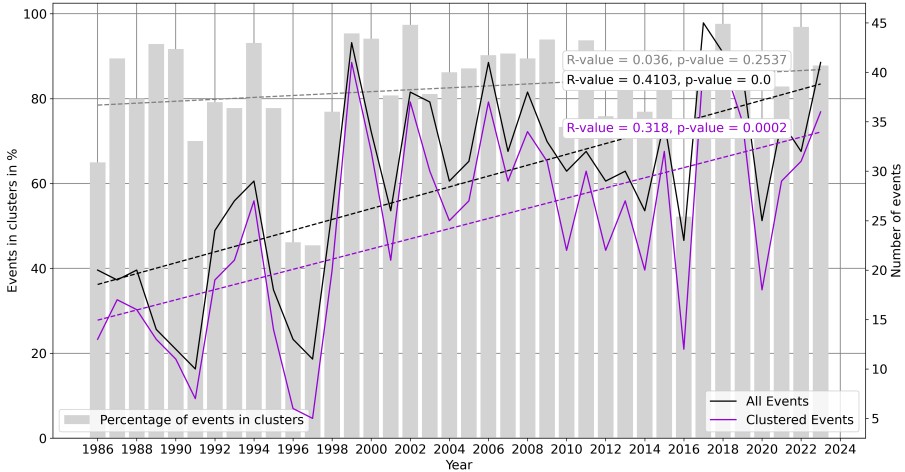

**Figure 11.** Time series of the percentage of meteorological hazards in clusters (clustering window = 14 days) as well as the number of all resp. clustered events during 1986–2023, including trend lines and significance.

mainly occurs during May–August for pluvial floods, convective gusts and hail, and during December–February for large-

scale storms as well as for fluvial floods. When events are defined using the 90th percentile of insured loss and claims and using the Peaks-over-Threshold method, clustering is significant for hail as well as for convective gust events from about 30 days during May–August (but not for pluvial or mixed floods). Clustering is also significant for large-scale storms and fluvial floods in winter compared to a random sample. This aligns with existing literature regarding the detection of clustering among extratropical cyclones (e. g., Dacre and Pinto, 2020) and the detection of no significant clustering regarding precipitation (e. g.,

Tuel and Martius, 2021a) in Europe. When two hazard types are combined, the degree of clustering is increased. Clustering is generally robust for the combination of three hazard types using a flexible event definition.

We have furthermore compared and evaluated different methods of declustering (= event definition) using a data-driven (Peaks-over-Threshold) vs. a predetermined (Hours Clause) method. We find that when using a fixed event definition, a significant deviation from a homogeneous Poisson process is detected in almost all cases. It should however be noted that events

have varying characteristics and resulting durations; the Hours Clause method does therefore not reflect their true occurrence.

We find a skewed distribution of losses, where a low number of events creates a large share of the overall losses. Nonetheless, clusters of convective resp. large-scale hazard types in summer resp. winter result in higher losses compared to their isolated occurrence. These clustered extremes have increased significantly throughout the past 38 years.

This study is unique regarding the use of impact data to assess clustering for a long time period from 1986–2023. However,

some limitations need to be taken into account: Insurance data are generally dependent on how claims are regulated. Although the losses are adjusted for inflation and the number of contracts, we cannot account for changes such as, e. g., policy adjustments or changes in exposed assets (e. g., solar panels on roofs), general wealth and the susceptibility to meteorological hazards (Kron et al., 2019). However, since the loss data do not show a significant increasing trend in annual losses, these factors might be



less relevant in this case. Furthermore, the damage regulation is biased towards the first day of the month, probably because of
simplicity to damage regulators; this however is within the scope of the usual fluctuation for the most extreme events. A bias of
insurance loss data to fraud is also possible; it is however assumed to be less relevant since we only evaluate major loss events.

The study is based on comprehensive data, but focuses on a limited geographic area. We therefore suggest to extend the
spatial scope in future studies. Furthermore, the impact we refer to is purely insurance-related and therefore monetary. Damage
to critical infrastructures or municipality assets are not captured by the data. Due to a lack of comparable data, no societal
impacts such as fatalities are included to measure impact. In addition, other hazard types such as cold spells, droughts or
heatwaves, which do not lead to a high direct monetary (insured) damage, were not included. These events also usually occur
on different timescales: Impact-relevant durations of those hazards range from about 2 weeks to two months (Polt et al., 2023)
and hydrological droughts cluster most strongly on the annual time scale and generally from seasonal to 3-year time scales
(Brunner and Stahl, 2023). Therefore, a comparison with the present hazard types with a mean duration of less than two days
is not sensible. It should furthermore not be neglected is that there is a stochastic element within impact data, which may lead
to the effect that a meteorologically relevant event at the local scale is not captured due to low population density and therefore
low losses. We argue that these events are less relevant to the public, since they do not create major damage.

Compound hazards are often observed related to specific atmospheric patterns such as atmospheric blocking (e.g., Kautz
et al., 2022). Future research could therefore be directed towards investigating the drivers of multivariate hazards, which, e. g.,
Bloomfield et al. (2024) did for Great Britain, where they connected daily flood-wind extremes to synoptic conditions. When
atmospheric patterns related to clusters of multivariate extremes are identified, future predictions of those hazards and their
joint occurrence could be enhanced. Clustering and atmospheric patterns have already been investigated regarding single hazard
types by Tuel and Martius (2022a, b); Yang and Villarini (2019); Villarini et al. (2011) for precipitation and by Vitolo et al.
(e.g., 2009) for extra-tropical cyclones. Another interesting topic for further research would be to investigate how the clustering
of different types of meteorological hazards changes due to climate change. This has been investigated for windstorms, where
Karwat et al. (2024) have shown that extratropical cyclone clustering is expected to increase significantly by 25 % in Europe
during 2060 to 2100.

We generally argue toward a holistic view of hazards, since a lot of research follows a single hazard type approach. Risk
can only be assessed accurately if we incorporate a multi-hazard view including all relevant types of hazards, interactions and
consequences.

*Data availability.* The insurance loss data are confidential and therefore not freely available. Both the daily precipitation data (HYRAS-DE-PRE) used to refine the flood category, and the hourly climate data used for subdividing the storm category are freely available from the Climate Data Center (CDC) of the DWD



*Author contributions.*   KK: Investigation, Formal analysis, Software, Visualization, Writing – original draft preparation, Writing – review &
editing; AT: Software, Writing – review & editing; MK: conceptualization, supervision, critical review of all drafts.

*Competing interests.*   The authors declare that they have no competing interests.

*Acknowledgements.*   Katharina Küpfer has been supported by a grant from the "Foundation for Protection Against Natural Hazards" (Stiftung
Umwelt und Schadenvorsorge). Katharina Küpfer would also like to thank Susanna Mohr and Achim Gegler for their support throughout
the project, and Markus Augenstein and Pierre Häsler for proofreading. Michael Kunz's participation in this work has been supported by the
Helmholtz Association (the Changing Earth – Sustaining our Future research programme).





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
