# Peer review of "Impact-based temporal clustering of multiple meteorological hazard types in southwestern Germany"

_EGUsphere, 2024_

## Author Comment (AC1)

**Authors' Response to Reviews of**

**Impact-based temporal clustering of multiple meteorological hazard types in southwestern Germany**

Katharina Küpfer, Alexandre Tuel, Michael Kunz

*Nat. Hazards Earth Syst. Sci.,* `https://doi.org/10.5194/egusphere-2024-2803`
* * *
RC: *Reviewers' Comment*,    AR: Authors' Response,    ☐ Manuscript Text

**Reviewer #1**

**General comments**

RC: *The paper analyzes the relevant question for risk assessment of the temporal clustering of events. The study is based on insurance loss data covering south-west Germany over the period 1986-2023. The hazard types associated to the losses are flood, storm and hail and they are associated to meteorological data. The hazards flood and storm are further separated into different phenomena since they can derive from different weather conditions: fluvial, pluvial or mixed floods on one hand and large-scale storms and convective gusts on the other hand, which makes sense. Then the methodology is described, and the results analyzed and discussed.*

AR:  We would like to thank Sylvie Parey for reviewing the manuscript and their valuable comments.

**Specific comments**

RC: *While seasonality is handled both in the loss data and in some hazards' characterization, it is not considered when identifying the major loss events. In my opinion, because seasonality is clearly identified in the loss distribution, it should be considered in the characterization, otherwise the percentile is computed in mixing different types of losses, which can be misleading. Therefore, I would suggest that the discussion devoted to the loss distribution analysis in section 4 is moved before the major loss events identification in section 2, justifying that the identification should be made on a seasonal basis. Then the same methodologies can be applied for each main season of occurrence and further summarized at the annual scale if necessary.*

AR:  We thank Sylvie Parey for raising this point, which we understand. As mentioned, the loss distribution shows strong seasonal patterns depending on the hazard type. However, by defining events depending on their seasonality, we would miss events and clusters that occur across the limits of seasons (e g., starting in late August and continuing until September, see Fig. 5). A seasonal event identification would also overly emphasize the relevance of, e. g., convective gusts during December–February. Conversely, if they were not considered at all because of lower losses or a low number of events, medium loss events of that type and season would be missed (e. g. fluvial floods during May–August). Actually, this annual definition of events allows us to see the seasonal pattern identified in Fig. 3. Off-season events or events across seasons can become relevant, for example, if they are also associated with clustering (see Fig. 5b). Events need to be defined before the seasonal analysis, as they are defined as meteorological event categories and not as seasonal categories.

Another reason for identifying events before the meteorological characterization is that the methods we use to

differentiate between hazards require a subset of extremes (e. g., convective gusts are identified by those events that do not occur within a large pressure gradient). Given the small-scale nature of these convective events, we are limited in our choice of methods of assigning meteorological categories.

Furthermore, if we moved the loss distribution analysis (Sect. 4.1) before the event identification (Sect. 2.2), the loss distribution analysis would consider daily losses instead of aggregated multi-day events. This would mean that events which actually occurred within the same system would be considered as separate events (see final comment for reviewer #1).

Finally, we agree that the percentile is computed mixing different types of losses, however these losses are registered as one category identified by the insurance company (storm vs. flood vs. hail event). Therefore, we would like to keep the order of the analysis and sections, but for further clarification, we have included a sentence in the manuscript after L163:

> Although all hazards follow strong seasonal patterns (see Sect. 4.2), percentiles are computed on an annual basis. This is because events and clusters can span over the limits of seasons, and because with an annual definition, the seasonal pattern within the loss data is kept.

**RC:** *The identified clustered hazards are physically relevant, which is reassuring, but one may wonder whether such an analysis was really necessary to derive the results. An interesting question regarding these events is the role of decadal variability, which is hard to infer with less than 40 years of observations. The identified clustering may be explained both by the fact that the clustered hazards derive from the same weather situation and by the fact that those weather situations occur more frequently during certain decades compared to others. This should be considered when analyzing trends too.*

AR: We thank Sylvie Parey for raising these points. We would like to split our answer in three parts.

1. **Relevance of clustering analysis vs. seasonal patterns:** Our results on the degree of clustering, as measured by Ripley's K, are somewhat surprising: We find significant clustering compared to random samples for the combination of extremes during 1986–2023, but not for all single hazards (Figs. 6–8). This cannot be identified from the mere occurrence of hazards (e. g. Fig. 3). To make this clearer, we have rearranged parts of the Abstract (L5-10):

> Results show that clustering is significant only for certain hazard types compared to a random time series. However, clustering is robust for a combination of multiple hazard types, namely hail, mixed or pluvial floods and storms. This particular combination of hazard types is also associated with higher losses compared to their isolated occurrence. Clusters of damaging hazards occur mainly during May–August and depend on the method of defining independent events (Peaks-over-Threshold with flexible lengths vs. Hours Clause with fixed lengths) and their resulting duration.

2. **Clustered hazards potentially deriving from the same weather situation:** To avoid that the identified clustering derives from the same weather situation, we performed a declustering with the Peaks-over-Threshold and Hours Clause methods (see Sect. 2.2). This declustering aggregates daily losses to multi-day events which ideally derive from the same weather situation. We have already included this in a detailed description in Sect. 2.2 (L163ff), to which we would like to add a sentence after L165 to make it clearer:

> Furthermore, as a prerequisite for applying extreme value statistics, the events are required to be independent. Toward this end, and to avoid clustering on the timescale of synoptic systems (around 5 days), clustering on the timescale of a few days needs to be removed (Wilks, 2006). This is called (runs) declustering (Coles, 2001) and means, in our case, that the daily data are aggregated to events with a length of either one or several days. Thereby, we avoid that events from the same synoptic cause appear as distinct events and lead to artificial clustering.

We acknowledge that in case of persistent patterns, several multi-day events may occur during the same weather situation. To test for robustness, we will therefore include a sensitivity test using the Peaks-over-Threshold method (POT) with $r = 3$, i. e., independent events defined when they are separated by less than 3 days. With this increase of days between events, we reduce the probability of events from the same weather situation to be wrongly considered as separate events. We expect little change in our results.

Furthermore, the average duration of an event is much higher using the HC method compared to POT (see L436f). This longer event duration, and therefore, also higher probability of independence, does not reduce the degree of clustering (see Figs. 6-8). This contradicts the argument of events being clustered due to the same physical causes. We would like to include this argument after L441:

> This furthermore proves that the (multi-day) events identified are not deriving from the same weather systems, since the degree of clustering is higher with longer durations (HC method) than with short ones (POT).

Lastly, a large part of the events during May–August (where our focus lies) is of convective nature, which is a short-lived phenomenon.

3. **Weather situation occurrence over decades:** We agree with Sylvie Parey that it would be interesting to analyse decadal variability, but also agree the time period is too short to perform an in-depth analysis. First, we would like to refer to Fig. 4 and the corresponding section for our discussion on inter-annual variability of event occurrence and to Fig. 11 and the corresponding section for our discussion on trends. To complement this discussion, we would like to include an analysis of the large-scale circulation by evaluating the event occurrence in relation to the North Atlantic Oscillation (NAO). We would like to discuss this in Sect. 5.5 and add two Figures in a supplement. As shown by Fig. 1 at the end of this document, events occurred most frequently during a negative NAO in May–August from 1986–2023. Particularly in recent years (since 2010), the NAO has been mainly negative during May–August. For events during December–February (see Fig. 2 at the end of this document), the opposite is the case. Synoptic storm events, which are most frequent during this season, occur predominantly within a positive NAO, which has become more frequent in recent years. This provides an additional explanation for the upward trend evaluated in Fig. 11.

**0.1. Technical corrections**

**RC:** *Technical corrections: Line 18: "Damage by those hazard": hazards; Line 98: the closing bracket should be removed after 36 000 km2; Line 110: "onlyf" is written instead of "only"; Line 126: "Given the the different environmental conditions" 2 instances of "the", one should be removed; line 472: "(see Fig 8" the closing bracket is missing; line 570: "It should furthermore not be neglected is that there is a stochastic element": "is" should be removed*

AR: We thank Sylvie Parey for the technical corrections; we addressed and implemented all of them in the main text.

[Figure]

Figure 1: May–August: (a) Distribution of monthly NAO values during 1986–2023, depending on the event type (colors), isolated occurrence or occurrence in clusters (hatches), (b) monthly mean NAO values from 1986–2023. Positive NAO values are detected when with mean values > 0.5 and max values > 0.75. Negative values relate to mean values < -0.5 and max values < -0.75. Neutral years are all years neither classified as positive nor as negative.

[Figure]

Figure 2: December–February: (a) Distribution of monthly NAO values during 1986–2023, depending on the event type (colors), isolated occurrence or occurrence in clusters (hatches), (b) monthly mean NAO values from 1986–2023. Positive NAO values are detected when with mean values > 0.5 and max values > 0.75. Negative values relate to mean values < -0.5 and max values < -0.75. Neutral years are all years neither classified as positive nor as negative.

---

## Author Comment (AC2)

**Authors' Response to Reviews of**

**Impact-based temporal clustering of multiple meteorological hazard types in southwestern Germany**

Katharina Küpfer, Alexandre Tuel, Michael Kunz
*Nat. Hazards Earth Syst. Sci.,* `https://doi.org/10.5194/egusphere-2024-2803`
* * *
**RC:** *Reviewers' Comment*,     AR: Authors' Response,     □ Manuscript Text

**Reviewer #3**

**General comments**

**RC:** *The study "Impact-based temporal clustering of multiple meteorological hazard types in southwestern Germany" is a detailed analysis of multi-hazard occurrence and related insured losses in Baden-Württemberg. I find the study well-written and methodologically sound.*

AR: We would like to thank Dominik Paprotny for reviewing this article and their valuable comments.

**RC:** *However, the main issue with the study is its relevance. As noted by the first reviewer, the analysed hazards are mostly clustered because they are physically connected to the same causes. This can be easily derived already from Figure 3 that the hazards are either caused by winter extra-tropical cyclones or summer convective storms. As the events are largely confined to two short seasons, analysing clustering up to 60 days will naturally show strong clustering.*

AR: For the part on potential clustering connected to the same physical causes, please see our answer to Reviewer #1, comment #2.

We agree that Fig. 3 shows a mainly seasonal occurrence and therefore a seasonal clustering. This is visible from Fig. 5a as well. However, Fig. 5b examines clusters forming due to the combination of different hazards, which can not be derived from Fig. 3. By using a clustering window of 21 days in Fig. 5, events occuring within the same season do not necessarily occur within a cluster.

On the other hand, we do analyse clustering up to 60 days using the statistical tool Ripley's K. We agree that due to the seasonal occurrence, an average event in a 'high season' will be surrounded by a relatively large number of events. We however disagree that this will naturally show strong clustering, since the degree of clustering is evaluated against a significance test of a random sample of (seasonal!) homogeneous Poisson processes with the same rate of occurrence. This is why the significance test is important, because it allows us to make statements on the degree of clustering, similarly done e. g. by Barton et al. (2016) or Tuel and Martius (2021). This is already described in detail in the main text (L227ff). We would like to complement this by the following paragraph:

> Note that this significance analysis actually determines the degree of clustering – due to seasonal patterns (see Sect. 4.2), events are likely to be surrounded by other events. The significance analysis serves the purpose of quantifying the deviation from the expected number of surrounding events and thereby provides

> an assessment of clustering.

If we analyse clustering on these seasonal timescales, seasonal clusters are often found (Figs. 6-8), but they require a different interpretation: Seasonal clustering here refers to some seasons showing a higher number of events while in other seasons, not many events are experienced. We already describe this in L225:

> Clustering on the seasonal level in our case compares the occurrence of events between different years (=seasons).

**RC:** *One question is how relevant are the losses of clustered events compared to the loss of the "main" event. Were the losses in the past 40 years really clustered, or single major events were strongly dominant and the co-occurring losses (of the same or different type) were not really important?*

AR: Firstly, we would like to refer to Fig. 2 for this point. As directly visible from the figure and stated e. g. in L305–312, the losses throughout 1986–2023 were dominated by a few main events. We also highlight (see Fig. 1) that clustering of the most severe loss events does not occur, but major events dominate the loss. As seen from Fig. 5, these major events are however also connected with other events, thus occurring in clusters (e. g., Lothar in 1999). With an increasing geographic scope, this is expected to become more relevant. In addition, smaller events can also lead to considerable damage, especially in the years with the absence of major events. We argue that those events with medium loss should be considered in relation to each other and not by hazard type. As stated in the next answer, insurance companies and the corresponding damage regulators may still be overburdened by one event, while the next event is already occurring.

However, we acknowledge the relevance of evaluating the losses of the main event vs. the total losses, since this is a common clustering metric applied to insurance data. We would like to address this point in Sect. 5.4, which we would like to rename to "Loss patterns and clustering", and complement this with an additional figure on the ratio of occurrence exceedance probability (OEP), i.e., the loss of the main event in a year, versus the annual exceedance probability (AEP), corresponding to the total loss in a year (see Fig. 3 at the end of the document). In contrast to other studies, OEP and AEP are applied to aggregate losses across hazards. It can be seen that in certain years (e. g., 1999, 2013), the ratio of OEP/AEP is very high, indicating a large contribution of a single event and thus a low degree of clustering. The mean ratio of OEP/AEP across 1986–2023 however equals 0.39, which means that on average, the contribution of several events is relevant to annual losses. It is also visible that the degree of clustering as measured by OEP/AEP is much higher during DJF than during MJJA. When evaluated against the return period, medium loss years never exceed an OEP/AEP ratio of 0.5.

**RC:** *I think the authors should more strongly highlight what they think is their contribution with this study, and especially why clustering is of any actual relevance in the study area in terms of societal and economic impacts, or at least for the insurance sector*

We thank Dominik Paprotny for this comment. We would like to split this answer in three parts:

1. **Economic and societal relevance of clustering:** Clustering is of actual relevance in the study area as described in the introduction, e. g. a succession of events in summer 2013 or in 2008. Economic impacts of clustering are evident from Fig. 10: Clustered events should not be neglected since clusters occur, and they lead to an increased loss compared to isolated events. This has also been found by Xoplaki et al. (2023) for Germany and by Hillier et al. (2015) for the UK. Examples of clustered hazards and their damage

amplification in Germany include an increased runoff in case of a heavy rain event after a heatwave, or an increased debris flow with damage potential in case of flooding after a storm event (see e. g., Kreibich et al. (2014)). Societal impacts include, for example, effects of cascading events (of also medium loss) while reparation is still ongoing: Authorities and technical organisations can become overburdened in a series of events due to limited availabilities for repeated action, since many of the organisations are based on volunteers. In longer periods of action, lack of food supply or rest of rescuers can harm recovery.

2. **Relevance for the insurance sector:** Likewise, insurance companies and their regulators can become be overburdened with a succession of several events. Since this depends on the number of major events (see Sect. 2.1.2), we would like to take up the suggestion to include an analysis of the ratio between maximum loss event and total seasonal loss (see previous answer and attached Fig. 1). We furthermore argue that insurance models should increasingly incorporate the view of interacting extremes, since we can see that they do not always occur independently. This is relevant considering our finding that statistically significant clustering can mainly be found if multiple hazards are analysed together. We would also like to point to the relevance of considering accurate event durations by comparing between the Hours Clause (HC) method and a Peaks-over-Threshold approach. Our results show that clustering depends on that event definition and that a fixed event duration (HC method) often overestimates the degree of clustering.

3. **Contribution to the research field:** We argue that our study contributes to the field by bridging between disciplines. For example, we make use of clustering metrics of primarily hydrologic data applied to impact data. By using impact data as a basis and refining it with meteorological data, we work across hazards and across the disciplines of meteorology, hydrology and impact research. This approach could be extended to other (non-meteorological) hazard types. An application regarding different types of impact, e. g. regarding capacities of authorities, would be possible as well.

We will make these points clearer in the main text (Introduction and Conclusions).

**RC:** *Other potentially major issue I see is indicated at the beginning of Sect. 2: "This study is based on loss data (Sect. 2.1.1) from a building insurance company". It is understandable that insurance data can't be shared publicly. However, at minimum, the insurance company needs to be identified by name, and in the data availability section, information needs to be provided how other researchers could apply to that company to also have access to that information. Making any verifiability of the study impossible is against the editorial policies: `https://www.natural-hazards-and-earth-system-sciences.net/policies/data_policy.html`*

AR: We thank Dominik Paprotny for raising this point. We have now included the name of the insurance company in the main text as well as in the data availability statement. We changed L91 and L105 to the following:

> This study is based on loss data (Sect. 2.1.1) from the SV Sparkassenversicherung building insurance company operating mainly in southwestern Germany and covers the period from 1986 to 2023.

> Extreme events, i.e. hydro-meteorological events leading to a major loss, are identified using data from the building insurance company SV Sparkassenversicherung.

as well as the first sentence in the data availability statement:

> The insurance loss data by the SV Sparkassenversicherung are confidential and therefore not freely available.

The underlying data of this analysis are confidential, since it is of interest to the insurance company to avoid that competitors draw conclusions from their data. They have unfortunately been made available for this project exclusively.

**RC:** *Finally, in L532: "Furthermore, this increasing trend is also influenced by non-meteorological factors, which could not be factored in." It's not true that it can't factored in. Exposure growth is major driver. Between 1991 and 2023, the value of fixed assets in Germany, in price-adjusted terms, increased by 68% for buildings and similar amount for other types of assets (as per Destatis database). Many studies on exposure-adjusted losses for different hazards are available. Also, strong exposure growth over the study period could affect event detection, creating an artificial upward trend in number of events.*

 AR: We thank Dominik Paprotny for pointing this out. As stated in Sect. 2.1.3, the data is 1) adjusted for inflation with a building-specific price index and 2) adjusted for the portfolio variability, which is a proxy for exposure, with the number of contracts. We agree that this does not reflect the market value of the buildings – however, when buildings are adjusted for building-specific inflation, this does reflect the repair costs. The (adjusted) losses should reflect those repair costs, while the market value of the building is unchanged. For example, if shutters become damaged by a hail event, this amounts to a different loss (or: reconstruction costs) today than it would have ten years ago. This difference in losses is reflected in the adjusted reinstatement costs, as reflected by the building price index and the corresponding adaptation factor. Only in very rare cases, buildings become complete write-offs after an extreme event, which would then also affect the market value, justifying an adjustment with the fixed asset values in Germany for buildings. We agree that exposure growth is a major driver of increased risk, but to our understanding, this additional correction is not relevant in our case, since we compare losses and not total market values, also in Fig. 11. Also, the loss history of a building is not directly related to its market value in Germany. We are happy to discuss that further if needed.

 However, we agree the sentence in L532 should be reformulated, since factors outside of meteorological conditions, but can partly not be included:

> This increasing trend is also influenced by non-meteorological factors, which are partly accounted for as described in Sect. 2.1.3, and could partly not be factored in, such as changed behavior of citizens, fluctuation in insurance regulation and a change in building vulnerability due to building materials.

**Minor comments**

**RC:** *L20: the statistic refers to Germany, not Europe. For losses in Europe, I would suggest referring to: `https://doi.org/10.2908/CLI_IAD_LOSS`*

 AR: We thank Dominik Paprotny for his thorough reading and for pointing that out. We are happy to include the proposed data source and cite it in summarized form the annual briefing of the EEA (`https://www.eea.europa.eu/publications/economic-losses-and-fatalities-from`). We furthermore would like to include a sentence on the losses in Germany with a new reference. The paragraph would then look as follows:

> Between 1980 and 2023, total losses caused by natural hazards are estimated at €738 billion in the European Union (European Environment Agency, 2023). Between 2001 and 2021, extreme events have amounted to annual losses of about 6.6 billion in Germany, including indirect effects ((Trenczek et al., 2022)). Flood and storm events are the major drivers of losses in Germany (Kreibich et al., 2014).

**RC:** *Section 2.2: It would be beneficial to know what is the magnitude of minimum and maximum losses of filtered events. Also, Figure 2 doesn't have any scale in the axes.*

 AR:  Unfortunately, we can neither include the scale on the axes of Fig. 2 nor the minimum and maximum losses of filtered events for confidentiality reasons. We argue that the main point of the manuscript is to analyze clustering, for which the absolute losses of the events do not matter.

**RC:** *Section 4.1: This section could be much shortened by moving the analysis why certain events are more damaging to the discussion (the last two paragraphs), because it is not the main topic of the paper.*

 AR:  We are happy to shorten the section. Some of the results on loss characteristics comparing different meteorological categories could, although not the main topic of the paper, be of general interest. Since, to our knowledge, this has not been published elsewhere, it can serve as an introduction to the topic of comparing between hazard types and their losses. Since we did not include a distinct discussion section, we would like to keep the order as it is.

**RC:** *Section 5.2: It would be good to explain to readers unfamiliar with it, that the 21-day window is also taken from insurance practice. Note that you explain this in context of specific 3- and 7-day windows.*

 AR:  This window is not initially taken from insurance practice but a window which corresponds to sub-seasonal clustering and hydrologically relevant durations (e. g., Barton et al. (2016), Kopp et al. (2021), Tuel and Martius (2021)). We will however include that 504 hours is the maximum hours clause we are aware of after L183. We would also like to include a paragraph explaining this choice when clustering is analysed after L382:

> Clustering is analysed on the timescale of 21 days since this is a common window for clustering analyses in the field of hydrology, since it includes hydrologically relevant durations. It furthermore is the longest time period of event identification with HC we are aware of (see Sect. 2.2.2).

**RC:** *Figure 11: Why do you suddenly switch to a 14-day window? Also, the labels shouldn't obscure the graph.*

 AR:  We thank the reviewer for pointing this out. We will change the graph for the 21-day window. This includes minor changes in the text. We furthermore changed the position of the graph labels so that they don't obscure the lines.

**References**

Yannick Barton, Paraskevi Giannakaki, Harald Von Waldow, Clément Chevalier, Stephan Pfahl, and Olivia Martius. Clustering of regional-scale extreme precipitation events in southern Switzerland. *Mon. Weather Rev.*, 144(1):347–369, 2016. . URL https://journals.ametsoc.org/view/journals/mwre/144/1/mwr-d-15-0205.1.xml.

J. K. Hillier, N. Macdonald, G. C. Leckebusch, and A. Stavrinides. Interactions between apparently 'primary'

[Figure]

[Figure]

[Figure]

Figure 1: Clustering measure of the main event contribution vs the overall loss: Occurrence Exceedance Probability, relating to the maximum loss event in a season, vs. the Aggregate Exceedance Probability, i.e. the total seasonal loss, during (a) May–August and (b) December–February

weather-driven hazards and their cost. *Environ. Res. Lett.*, 10(10):104003, September 2015. ISSN 1748-9326. .

Jérôme Kopp, Pauline Rivoire, S Mubashshir Ali, Yannick Barton, and Olivia Martius. A novel method to identify sub-seasonal clustering episodes of extreme precipitation events and their contributions to large accumulation periods. *Hydrol. Earth Syst. Sci*, 25(9):5153–5174, 2021.

Heidi Kreibich, Philip Bubeck, Michael Kunz, Holger Mahlke, Stefano Parolai, Bijan Khazai, James Daniell, Tobia Lakes, and Kai Schröter. A review of multiple natural hazards and risks in Germany. *Nat. Hazards*, 74:1–26, June 2014. . URL https://link.springer.com/article/10.1007/s11069-014-1265-6.

Jan Trenczek, Oliver Lühr, Lukas Eiserbeck, Myrna Sandhövel, and Viktoria Leuschner. Übersicht vergangener extremwetterschäden in deutschland. Technical report, 2022.

Alexandre Tuel and Olivia Martius. A global perspective on the sub-seasonal clustering of precipitation extremes. *Wea. Clim. Extrem.*, 33:100348, September 2021. ISSN 2212-0947. . URL https://www.sciencedirect.com/science/article/pii/S2212094721000426.

Elena Xoplaki, Florian Ellsäßer, Jens Grieger, Katrin M. Nissen, Joaquim Pinto, Markus Augenstein, Ting-Chen Chen, Hendrik Feldmann, Petra Friederichs, Daniel Gliksman, Laura Goulier, Karsten Haustein, Jens Heinke, Lisa Jach, Florian Knutzen, Stefan Kollet, Jürg Luterbacher, Niklas Luther, Susanna Mohr, Christoph Mudersbach, Christoph Müller, Efi Rousi, Felix Simon, Laura Suarez-Gutierrez, Svenja Szemkus, Sara M. Vallejo-Bernal, Odysseas Vlachopoulos, and Frederik Wolf. Compound events in Germany in 2018: drivers and case studies. *EGUsphere*, pages 1–43, August 2023. . URL https://egusphere.copernicus.org/preprints/2023/egusphere-2023-1460/.

---

## Author Comment (AC3)

**Authors' Response to Reviews of**

**Impact-based temporal clustering of multiple meteorological hazard types in southwestern Germany**

Katharina Küpfer, Alexandre Tuel, Michael Kunz
*Nat. Hazards Earth Syst. Sci.,* `https://doi.org/10.5194/egusphere-2024-2803`
* * *
RC: *Reviewers' Comment*,    AR: Authors' Response,    ☐ Manuscript Text

**Reviewer #2**

**General comments**

RC: *The study examines how clusters of extreme weather events in southwestern Germany lead to higher losses than random occurrences. They analyzed insurance loss data from 1986 to 2023, adjusted for inflation and contract changes, using algorithms and clustering metrics to assess the significance of clustering for different hazards and their combinations. The research shows that clusters, particularly involving hail, floods, and storms, mainly happen in the summer and are linked to increased losses. The study highlights the growing number of extreme weather clusters over 38 years and their impact on risk assessment and insurance. The authors advocate for a holistic approach to hazard and risk analysis, considering the amplified risks of multiple combined hazards, especially in the context of climate change.*

AR: We would like to thank anonymus reviewer #2 for reviewing this manuscript and their valuable ideas and comments.

RC: *The work is of great relevance for the multi-hazard community and makes a step forward towards the understanding of complex risk dynamics. I would recommend the publication of the manuscript if the authors are able to further clarify or justify the following points: 1. I understand the data cannot be made publicly available but the authors should at least share the code so that a reader can follow the detailed steps of the analysis and perhaps reproduce it with different datasets.*

AR: We thank the anonymous reviewer for the comment. We are happy to share the code used and will provide it with the final manuscript.

RC: *2. The clustering for both single and multi hazard only takes into account the temporal dimension. Wouldn't it be better to do a spatio-temporal clustering especially considering that the weather data comes at 1 km2 resolution? How the results would change?*

AR: We agree that a spatio-temporal clustering would also be interesting to analyse. However, the underlying insurance dataset is only available in a spatially aggregated form, i. e., for the whole domain of Baden-Württemberg. Therefore, we cannot perform a spatio-temporal clustering analysis. Furthermore, we are interested in aggregated impacts over the domain of Baden-Württemberg with a limited extent of $36\,000$ km$^2$. Please see the following sentence (L112-114):

> As there is no finer spatial information (such as e.g. on the municipality level) available in our dataset, we use the spatially aggregated losses per day for the whole region. Due to the limited size of BW, we can assume that there is only a single synoptic cause of major events at the same time.

**RC:** *3. Can we assume that the percentage of insured assets is constant across the considered region and over time? How would this influence the results of the analysis?*

AR: The percentage of insured assets fluctuates across time. This is taken into account by adjusting for the number of contracts the insurance company holds (L146ff, slightly modified):

> The portfolio variability is adjusted as follows: following the abolition of the insurance obligation in 1994 in BW, the portfolio has declined almost continuously. We therefore additionally adjust the insured losses with a factor that captures the number of contracts (NC) in the course of time, where $NC_{\text{mean}}$ refers to the mean contract number over the entire time period [...]

We do unfortunately not have any recent information on the regional variability of the portfolio since the data is only available for the spatial resolution of BW (see previous answer). However, in past data from the insurance company with a higher spatial resolution, the regional variability was negligible (not published). We will therefore include a sentence in the manuscript after L156:

> The regional variability is assumed to be uniform due to past analyses of this data in a higher spatial resolution (not published).

Since we are not investigating regional differences, we therefore assume a uniform insurance coverage across the geographic scope and take into account temporal variability by adjusting with the number of contracts.

**RC:** *4. It would be interesting to add the vulnerability dimension in the study since the loss declared would depend also on that and not just the hazard number.*

AR: We agree that it would be interesting to add the vulnerability dimension in the study. Vulnerability functions in insurance models are usually depicted using, e. g., inundation depth in the case of flooding or hail size for a hail model, both compared to the damage. To determine these functions, object or at least georeferenced data would be required. Unfortunately, for this region, data is not available in such a fine spatial resolution. Furthermore, our goal is not to assess the risk, but to determine clustering of events based on insured losses.

**RC:** *5. Why not trying a loss independent clustering too? e.g. based on hazard intensity or other parameters.*

AR: We agree that this is an interesting point. As impact-based clustering is the main goal of this study and there is already vast literature on clustering independent of impacts (summarized in L55-59), we are afraid that we cannot take this suggestion. Furthermore, we expect robustness to the choice of input data by taking a relatively low percentile (90th percentile), a high population density in BW and a high coverage by the SV SparkassenVersicherung. For these reasons, we assume that a large part of all hazards of the types considered are impact-related and included within the adjusted insurance dataset. We would like to include these arguments in a modified paragraph in Sect. 5.3 (L495ff):

> To our knowledge, there are no other studies quantifying the degree of temporal clustering with respect to different types of (meteorological) hazards. We therefore contribute to the literature by considering a variety of meteorological hazard types related to impact data and finding that they do cluster when combined, irrespective of the event definition. We expect that these results are robust with regard to the choice of input data for this region, since we can assume that a large part of all major natural hazards are included in our impact datasets for the following reasons: Firstly, population density is generally high in Germany, and exceeded 100 inhabitants/$km^2$ for all districts in Baden-Württemberg as of 2022 (Statistisches Landesamt Baden-Württemberg, 2022). Secondly, insurance coverage against all hazards included in this analysis is very high across BW, and the SV Sparkassenversicherung has a high market share (see Sect. 2.2.1). Finally, by using the 90th percentile across all years, we include a large number of events (see e g., Fig. 4). Thereby, a major part of meteorological hazards, also in less densely populated regions with lower insured losses, should be included as well.

Additionally, we included a statement in the Conclusions after L572 (last sentence in this quote):

> It should furthermore not be neglected that there is a stochastic element within impact data, which may lead to the effect that a meteorologically relevant event at the local scale is not captured due to low population density and therefore low losses. We argue that these events are less relevant to the public, since they do not create major damage. Nevertheless, future research could be directed at analysing these clustering patterns with larger datasets and including larger geographic scopes.

**RC:** *6. I would also discuss more why daily insured damage. At a first glance it seems too short as a temporal window and prone to errors, double counting and so on.*

AR:  Agreed, daily insured damage could lead to errors, that is why we aggregate events to either single-day or multi-day events using the Peaks-over-Threshold and the Hours Clause method. This is a way of declustering, which removes double counting (see Sect. 2.2, L165–166). This is already discussed (L435-441).

**RC:** *7. How is exposure taken into account? Is it assumed to be uniform over the whole region? Somewhat related to question 3. & 4.*

AR:  Please see our answer to reviewer #3, comment #6 as well as our answers to comments #3 and #4 by reviewer #2.

**RC:** *I would like the authors to discuss further these points in the main text or at least provide an explanation behind their choices. Having said that, I acknowledge the relevance as well as the technical robustness of the work done by the authors.*

AR:  We would like to thank reviewer #2 again for their comments. Above, we stated for each comment whether we included a discussion in the text or explained our choices in the author response.